# Lecture notes on large deviations in non-equilibrium diffusive systems

**Bernard Derrida**

Collège de France, 11 place Marcelin Berthelot, 75005 Paris, France
and
Laboratoire de Physique de l'Ecole Normale Supérieure, ENS, Université PSL, CNRS,
Sorbonne Université, Université de Paris, F-75005 Paris, France

derrida@phys.ens.fr
June 6, 2025

## Abstract

These notes are a written version of lectures given in the 2024 Les Houches Summer School on *Large deviations and applications*. They are are based on a series of works published over the last 25 years on steady properties of non-equilibrium systems in contact with several heat baths at different temperatures or several reservoirs of particles at different densities. After recalling some classical tools to study non-equilibrium steady states, such as the use of tilted matrices, the Fluctuation theorem, the determination of transport coefficients, the Einstein relations or fluctuating hydrodynamics, they describe some of the basic ideas of the macroscopic fluctuation theory allowing to determine the large deviation functions of the density and of the current of diffusive systems.

# 1   Equilibrium versus non equilibrium

The main physical situations discussed in these lectures are steady states of systems in contact with one or several heat baths or one or several reservoirs of particles. Most of the systems considered here are finite, in the sense that the number of particles or the amount of energy that a system can absorb is finite. We will also assume that, by waiting long enough, the system reaches a steady state independent of its initial condition.

## 1.1   Equilibrium

For a system in contact with a single heat bath at temperature $T$, the steady state is nothing but the equilibrium. Then according to the canonical ensemble, the probability of finding the system in a microscopic configuration $\mathcal{C}$ is given by

$$\mathbb{P}_{\text{equilibrium}}(\mathcal{C}) = \frac{e^{-\beta E(\mathcal{C})}}{Z(\beta)}, \quad \text{with} \quad \beta = \frac{1}{k_B T} \tag{1}$$

where $E(\mathcal{C})$ is the energy of the configuration $\mathcal{C}$ and the normalization factor $Z(\beta)$ is the partition function. (In the following we will often use units where the Boltzmann constant $k_B = 1$.)

For example, for a classical gas of $N$ mono-atomic particles having mass $m$ with pair interactions between the particles, a microscopic configuration $\mathcal{C}$ is specified by the position $\vec{q}_i$ and the momentum $\vec{p}_i$ of each particle

$$\mathcal{C} = \left\{ \vec{q}_1, \ldots, \vec{q}_N; \vec{p}_1, \ldots, \vec{p}_N \right\} \quad . \tag{2}$$

with an energy $E(\mathcal{C})$ given by

$$E(\mathcal{C}) = \sum_{i=1}^{N} \frac{\vec{p}_i^{\,2}}{2m} + \sum_{i \neq j} V(\vec{q}_i - \vec{q}_j) \tag{3}$$

where $V(\vec{r})$ is the interaction energy between a pair of particles at distance $\vec{r}$.

Another example would be a lattice gas on a lattice of $L$ sites. In this case a microscopic configuration $\mathcal{C}$ is specified by the number $n_i$ of particles on each site $i$

$$\mathcal{C} = \{n_1, n_2, \ldots, n_L\} \tag{4}$$

and its energy $E(\mathcal{C})$ by a function of these occupation numbers

$$E(\mathcal{C}) = E(\{n_i\}) \quad .$$

At equilibrium, one usually does not need to care about the nature of the contact with the heat bath and one can start from (1) to try to derive the equilibrium properties of the system.

## 1.2   Non equilibrium steady state

For a system in contact with two heat baths at different temperatures $T_1$ and $T_2$, as in figure 1, there is usually no expression of the $P_{\text{steady state}}(\mathcal{C})$ which generalizes the Gibbs measure (1) because the knowledge of $T_1$ and $T_2$ is not sufficient.

$$P_{\text{steady state}}(\mathcal{C}) \; = ? \tag{5}$$

In particular, unlike in the equilibrium case (1), $P_{\text{steady state}}(\mathcal{C})$ depends on the nature of the

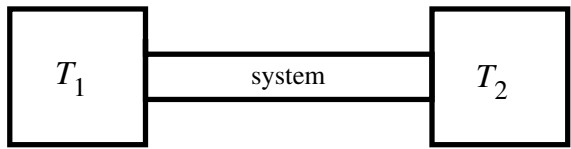

Figure 1: A system in contact with two heat baths at different temperatures

contacts with the heat baths. For example having a weak contact with the left heat bath at temperature $T_1$ makes the system heavily influenced only by the right heat bath and have $\mathbb{P}(\mathcal{C}) \simeq \mathbb{P}_{\text{equilibrium at } T_2}(\mathcal{C})$. On the other hand if the contact with the right heat bath at temperature $T_2$ is weak, one has $\mathbb{P}(\mathcal{C}) \simeq \mathbb{P}_{\text{equilibrium at } T_1}(\mathcal{C})$.

It is then clear that one needs to represent the effect of heat baths on the system.

One needs to represent the heat baths

## 2   Description of heat baths

Let us now see how one can represent the coupling of a system with heat baths [136].

### 2.1   Deterministic heat baths with deterministic dynamics

One possibility is to use deterministic heat baths. Their effect is to modify the dynamics of the particles in contact with the heat baths. There are several ways to introduce deterministic thermostats [89, 136]. One of them is called the Gaussian thermostat. For a one dimensional classical system evolving according to Newton's dynamics, the Gaussian thermostat consists in keeping the usual Newton equation for all the particles $i$ which are not in direct contact with the thermostat

$$m\ddot{q}_i = F_i \qquad \text{if } i \text{ is not in contact with the thermostat} \tag{6}$$

and in replacing it by

$$m\,\ddot{q}_j = F_j - \alpha\,\dot{q}_j \qquad \text{if } j \text{ is in contact with the thermostat} \tag{7}$$

for all particles $j$ in contact with a thermostat at temperature $T$, the parameter $\alpha$ being chosen such that the total kinetic energy of all the particles $j$ in contact with the heat bath remains fixed i.e.

$$\sum_{j \text{ in contact}} \left[ \frac{m\,\dot{q}_j^2}{2} - \frac{k_B T}{2} \right] = 0 \tag{8}$$

i.e.

$$\alpha = \left[ \sum_{j \text{ in contact}} \dot{q}_j\, F_j \right] \Big/ \left[ \sum_{j \text{ in contact}} \dot{q}_j^{\,2} \right] \quad . \tag{9}$$

in the one dimensional case, choosing in in initial condition satisfying (8).

Clearly this can be generalized to systems in contact with several heat baths, with some particles $j$ being in contact with one heat bath at a certain temperature $T_1$ and some other particles $j'$ in contact with another heat bath at temperature $T_2$: for the particles connected to each heat bath, one can impose a constraint like (8) so that there is an $\alpha_i$ associated to each thermostat.

Because this modified dynamics is still deterministic, each configuration $\mathcal{C}_t$ depends only on the initial condition $\mathcal{C}_0$

$$\mathcal{C}_t = \mathbb{F}_t(\mathcal{C}_0) \tag{10}$$

and the probability $\mathbb{P}_t(\mathcal{C})$ of finding the system in configuration $\mathcal{C}$ at time $t$ is given by

$$\mathbb{P}_t(\mathcal{C}) = \int \delta\Big(\mathcal{C} - \mathbb{F}_t(\mathcal{C}_0)\Big)\, \mathbb{P}_0(\mathcal{C}_0)\, d\mathcal{C}_0 \quad. \tag{11}$$

For a deterministic system, the fluctuations and the large deviations are only due to the choice of the initial condition $\mathcal{C}_0$. Determining then the large deviation function of the empirical measure is usually not an easy task even for some simple dynamical systems (see for example the case of the one dimensional logistic map [124].)

## 2.2 Stochastic heat baths with Langevin dynamics

A very common way of representing the action of heat baths on particles is to add a Langevin force by replacing the Newton equation by a stochastic differential equation

$$m\ddot{q}_i = F_i - \gamma_i\, \dot{q}_i + \eta_i(t) \tag{12}$$

where $\gamma_i$ represents the strength of the contact of particle $i$ with a heat bath at temperature $T_i$ and $\eta_i(t)$ is a Gaussian white noise in time characterized by the following covariance

$$\langle \eta_i(t)\, \eta_i(t') \rangle = 2\gamma_i\, k_B\, T_i\, \delta(t - t') \quad. \tag{13}$$

Now the configuration $\mathcal{C}_t$ of the system at time $t$ depends not only on the initial configuration $\mathcal{C}_0$ but also on the effect of the noise

$$\text{Noise}_{0 \to t} \equiv \{\eta_i(t'), 0 < t' < t\}$$

i.e. on the stochastic forces which represent the action of the heat baths on the system

$$\mathcal{C}_t = \mathbb{F}_t(\mathcal{C}_0, \text{Noise}_{0 \to t}) \,. \tag{14}$$

The probability $\mathbb{P}_t(\mathcal{C})$ of finding the system in configuration $\mathcal{C}$ at time $t$ becomes

$$\mathbb{P}_t(\mathcal{C}) = \int \mathbb{P}_0(\mathcal{C}_0)\, d\mathcal{C}_0 \int \rho(\text{Noise}_{0 \to t})\, d\text{Noise}_{0 \to t}\ \ \delta\Big(\mathcal{C} - \mathbb{F}_t(\mathcal{C}_0, \text{Noise}_{0 \to t})\Big) \tag{15}$$

and from (12) one can show that its evolution is given by

$$\frac{d\mathbb{P}_t(\{q_i, \dot{q}_i\})}{dt} = \sum_i \left[ -\dot{q}_i \frac{\partial\, \mathbb{P}_t}{\partial\, q_i} + \frac{\gamma_i}{m} \frac{\partial(\dot{q}_i\, \mathbb{P}_t)}{\partial\, \dot{q}_i} - \frac{F_i}{m} \frac{\partial\, \mathbb{P}_t}{\partial\, \dot{q}_i} + \frac{\gamma_i\, k_B\, T_i}{m^2} \frac{\partial^2\, \mathbb{P}_t}{\partial\, \dot{q}_i{}^2} \right] \quad. \tag{16}$$

It is easy to check that this modified dynamics leaves the canonical distribution (1,3) unchanged when all the $T_i$ are equal to $T$ and

$$F_i = -\sum_{j \neq i} \frac{\partial\, V(q_i - q_j)}{\partial\, q_i} \,.$$

On the other hand, it is clear that one can couple this way different particles to heat baths at different temperatures. In this case, the steady states correspond to the distributions invariant under the evolution (15,16), i.e. to the solutions of

$$\mathbb{P}(\mathcal{C}) = \int \mathbb{P}(\mathcal{C}_0)\, d\mathcal{C}_0 \int \rho(\text{Noise}_{0 \to t})\, d\text{Noise}_{0 \to t}\ \ \delta\Big(\mathcal{C} - \mathbb{F}_t(\mathcal{C}_0, \text{Noise}_{0 \to t})\Big)\ \ . \qquad (17)$$

## 2.3 Markov process

One often uses a Markov process to describe the exchanges between a system and the heat baths, in particular when the variables which characterize the configurations are discrete, (as in the case of a lattice gas where the variables are the numbers $n_i$ of particles on site $i$). One can then choose either a discrete time or a continuous time Markov process for the evolution of the system.

Discrete time Markov process

For a discrete time dynamics, the evolution of the system is specified by a sequence of configurations $\mathcal{C}_0, \mathcal{C}_1, \ldots, \mathcal{C}_t$ indexed by the time step $t$. This sequence of configurations is generated by a Markov process if the probability of being in the configuration $\mathcal{C}_{t+1}$ at time $t+1$ depends only on the configuration $\mathcal{C}_t$ at the previous time step and this probability is given by a matrix $M(\mathcal{C}_{t+1}, \mathcal{C}_t)$. All the matrix elements $M(\mathcal{C}', \mathcal{C}) \geq 0$ and they satisfy

$$\sum_{\mathcal{C}'} M(\mathcal{C}', \mathcal{C}) = 1 \quad \text{for all}\ \ \mathcal{C}\ \ . \qquad (18)$$

The probability $\mathbb{P}_t(\mathcal{C})$ evolves according to the following Master equation

$$\mathbb{P}_{t+1}(\mathcal{C}) = \sum_{\mathcal{C}'} M(\mathcal{C}, \mathcal{C}')\, \mathbb{P}_t(\mathcal{C}') \qquad (19)$$

and this implies that

$$\mathbb{P}_t(\mathcal{C}) = \sum_{\mathcal{C}_0} M^t(\mathcal{C}, \mathcal{C}_0)\, \mathbb{P}_0(\mathcal{C}_0)\ \ . \qquad (20)$$

Continuous time Markov process

For a Markov process with a continuous time, the probability for the system to jump from configuration $\mathcal{C}$ to a configuration $\mathcal{C}'$ during an infinitesimal time interval $dt \ll 1$ is given by $M(\mathcal{C}', \mathcal{C})\, dt$ and the evolution of $\mathbb{P}_t(\mathcal{C})$ becomes

$$\frac{d\mathbb{P}_t(\mathcal{C}')}{dt} = \sum_{\mathcal{C}'} M(\mathcal{C}, \mathcal{C}')\, \mathbb{P}_t(\mathcal{C}')$$

where the normalization condition (18) is replaced by

$$M(\mathcal{C}, \mathcal{C}) = -\sum_{\mathcal{C}' \neq \mathcal{C}} M(\mathcal{C}', \mathcal{C}) \quad \text{for all}\ \ \mathcal{C}\ \ . \qquad (21)$$

A continuous Markov process can be viewed as a limiting case of a discrete Markov process with transition matrix $\delta(\mathcal{C}', \mathcal{C}) + M(\mathcal{C}', \mathcal{C})\, dt$ in the limit $dt \to 0$. Note that the Langevin dynamics (12) can also be viewed as a limiting case of a continuous time Markov process.

## 2.4  Detailed balance

One way to ensure that the canonical distribution (1) at temperature $T$ is stationary for a (continuous or discrete time) Markov process is to require that the transition rates satisfy the detailed balance condition

$$M(\mathcal{C}',\mathcal{C}) \, \exp\left(-\beta E(\mathcal{C})\right) \, = \, M(\mathcal{C},\mathcal{C}') \, \exp\left(-\beta E(\mathcal{C}')\right) \tag{22}$$

where $E(\mathcal{C})$ is the energy of configuration $\mathcal{C}$.

A simple case where this detailed balance condition is not satisfied would be, in the continuous time case, a Markov matrix of the form

$$M(\mathcal{C}',\mathcal{C}) = M_1(\mathcal{C}',\mathcal{C}) + M_2(\mathcal{C}',\mathcal{C}) \tag{23}$$

where the Markov matrices $M_1$ and $M_2$ satisfy detailed balance conditions at different temperatures i.e.

$$M_i(\mathcal{C}',\mathcal{C}) \, \exp\left(-\beta_i E(\mathcal{C})\right) \, = \, M_i(\mathcal{C},\mathcal{C}') \, \exp\left(-\beta_i E(\mathcal{C}')\right) \tag{24}$$

with $\beta_1 \neq \beta_2$. This could represent a system in contact with two heat baths.

## 2.5  The symmetric simple exclusion process

The one-dimensional Symmetric Simple Exclusion Process (SSEP) [104,105,125,127] can be viewed as a system in contact with two heat baths. It describes a one dimensional lattice of $L$ sites. Each site is either empty or occupied by a single particle. Each particle hops to each of its neighbors at rate 1 if the target site is empty. Otherwise it does not move. (SSEP: Symmetric because hopping rates to the left or to the right are the same, Simple because, in a single step, a particle is only allowed to jump to one of its nearest neighbors either to its left or to its right, Exclusion because a particle cannot jump to an already occupied site). Moreover, as in figure 2, a particle can enter the system into the first site on the left at rate $\alpha$ and exit from this site at rate $\gamma$. Similarly, on the last site on the right, a particle can enter at rate $\delta$ or exit at rate $\beta$. As each site can be either occupied or empty, the total number of configurations $\mathcal{C}$ allowed for such a system is $2^L$.

One can think of each occupied site as being occupied by a particle or by a quantum of energy. The SSEP can therefore be viewed as a model of out-of-equilibrium transport, where quantities (particles, energy...) cannot accumulate indefinitely in the system and flow in and out of it.

Let us see whether the dynamics of the SSEP satisfy the detailed balance condition (22).

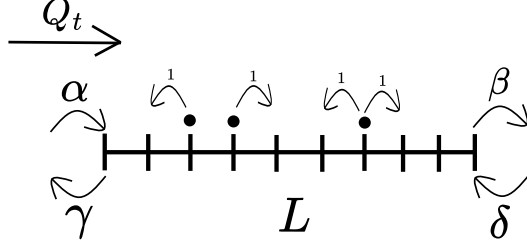

Figure 2: SSEP on $L$ sites with open boundary conditions

Because all the internal moves occur at rate 1, whatever the temperature, all configurations with the same number $n$ of occupied sites must have the same energy $E_n$. Now if one wants the injection and the exit rates on site 1 to satisfy detailed balance at a certain temperature $T_1$, one should have

$$\alpha \exp\left(-\beta_1 E_n\right) = \gamma \exp\left(-\beta_1 E_{n+1}\right) \quad . \tag{25}$$

Similarly, at the right boundary, a detailed balance condition would correspond to a temperature $T_2$ such that

$$\delta \exp\left(-\beta_2 E_n\right) = \beta \exp\left(-\beta_2 E_{n+1}\right) \quad . \tag{26}$$

We see that, unless the boundary rates satisfy $\alpha\beta = \gamma\delta$, the temperatures $T_1$ and $T_2$ are different and the Markov process which represents the dynamics of the SSEP does not satisfy detailed balance at a single temperature. On the other hand one write

$$M = M_0 + M_1 + M_2$$

where $M_0$ represents the internal moves (which keep the energy unchanged) and $M_1$ and $M_2$ represent the exchanges of energy with the heat baths at the left and the right boundaries. So, unless $\alpha\beta = \gamma\delta$, the SSEP is a model of a system in contact with two heat baths at different temperatures.

## 3 Large deviations of current

For Markov processes, tilted matrices matrices play a central role in the understanding of large deviations [28, 50, 59, 60, 101, 102, 131, 132].

### 3.1 The tilted matrix

One basic quantity one may study for the steady state of a system in contact with two or more heat baths is the flux of energy $Q_t$ through the system during time $t$. In the previous example of the SSEP, one can define this flux $Q_t$ as the total number of particles entering the system from the left minus the number of particles leaving at the left boundary during time $t$ (see Figure 2).

$$Q_t = (\text{no. of particles entered from left up to } t) - (\text{no. of particles exit to the left up to } t) \tag{27}$$

For a discrete time Markov process (remember that a continuous time Markov process is a limiting case of a discrete Markov process) one has

$$Q_t = \sum_{\tau=0}^{t-1} q(\mathcal{C}_{\tau+1}, \mathcal{C}_\tau) \tag{28}$$

where $q(\mathcal{C}', \mathcal{C})$, in the example of the SSEP, can take three possible values

$$q(\mathcal{C}', \mathcal{C}) \equiv q(\mathcal{C}' \leftarrow \mathcal{C}) = \begin{cases} +1 & \text{if a particle enters site 1 from the left} \\ -1 & \text{if a particle exits site 1 to the left} \\ 0 & \text{otherwise} \end{cases} \tag{29}$$

(For more general models, $q(\mathcal{C}', \mathcal{C})$ is not limited to $-1, 0$ or $+1$ values but it can take arbitrary integer or real values).

Let $\mathbb{P}_t(\mathcal{C}, Q)$ be the probability that $\mathcal{C}_t = \mathcal{C}$ and $Q_t = Q$ at time $t$. Its evolution is given by the following Master equation

$$\mathbb{P}_{t+1}(\mathcal{C}, Q) = \sum_{\mathcal{C}} M(\mathcal{C}, \mathcal{C}') \mathbb{P}_t\left(\mathcal{C}', Q - q(\mathcal{C}, \mathcal{C}')\right) \tag{30}$$

where $\mathcal{C}$ are the $2^L$ configurations of the system, and $Q_t \in \mathbb{Z}$.

If one introduces the following generating function of the integrated current $Q_t$

$$\widehat{\mathbb{P}}_t(\mathcal{C}) = \sum_{Q \in \mathbb{Z}} e^{\lambda Q} \mathbb{P}_t(\mathcal{C}, Q) \tag{31}$$

one can easily show that its satisfies

$$\widehat{\mathbb{P}}_{t+1}(\mathcal{C}) = \sum_{\mathcal{C}'} M_\lambda(\mathcal{C}, \mathcal{C}')\widehat{\mathbb{P}}_t(\mathcal{C}') \implies \widehat{\mathbb{P}}_t = M_\lambda^t\, \widehat{\mathbb{P}}_0 \tag{32}$$

with the tilted matrix $M_\lambda$ given by

$$M_\lambda(\mathcal{C}, \mathcal{C}') = e^{\lambda q(\mathcal{C}, \mathcal{C}')} M(\mathcal{C}, \mathcal{C}') \quad . \tag{33}$$

This yields the following large-$t$ behavior, up to non-exponential prefactors

$$\widehat{P}_t(\mathcal{C}) \sim e^{t\mu(\lambda)} \tag{34}$$

where $e^{\mu(\lambda)}$ is the largest eigenvalue of the tilted matrix $M_\lambda$.

The same holds for a continuous time Markov process with (32) being replaced by

$$\frac{d\widehat{\mathbb{P}}_t(\mathcal{C})}{dt} = \sum_{\mathcal{C}'} M_\lambda(\mathcal{C}, \mathcal{C}')\widehat{\mathbb{P}}_t(\mathcal{C}') \implies \widehat{\mathbb{P}}_t = e^{tM_\lambda}\, \widehat{\mathbb{P}}_0 \tag{35}$$

and $\mu(\lambda)$ being now the largest eigenvalue of the matrix $M_\lambda$.

**Remark:** since by definition (see (31,34))

$$\mu(\lambda) = \lim_{t\to\infty} \frac{1}{t}\log\langle e^{\lambda Q_t}\rangle \quad ,$$

one can determine from the knowledge of $\mu(\lambda)$ all the cumulants of the current in the long time limit

$$\lim\frac{\langle Q_t\rangle}{t} = \mu'(0) \quad ; \quad \lim\frac{\langle Q_t^2\rangle - \langle Q_t\rangle^2}{t} = \mu''(0) \quad ; \quad \lim\frac{\langle Q_t^n\rangle_c}{t} = \mu^{(n)}(0) \quad . \tag{36}$$

### 3.2 Example: SSEP with $L = 1$ (Quantum dot)

As an example let us consider the SSEP of figure 2 in the simple case $L = 1$. We denote $\mathbb{P}_0$ the probability of finding the system in the empty state and $\mathbb{P}_1$ for the occupied state so that the Master equation is

$$\begin{aligned}\frac{d\mathbb{P}_1}{dt} &= -(\beta + \gamma)\mathbb{P}_1 + (\alpha + \delta)\mathbb{P}_0 \\ \frac{d\mathbb{P}_0}{dt} &= (\beta + \gamma)\mathbb{P}_1 - (\alpha + \delta)\mathbb{P}_0 \quad .\end{aligned} \tag{37}$$

The Markov matrix $M$ is of the form

$$M = M_1 + M_2 \tag{38}$$

where

$$M_1 = \begin{pmatrix} -\gamma & \alpha \\ \gamma & -\alpha \end{pmatrix} \qquad M_2 = \begin{pmatrix} -\beta & \gamma \\ \beta & -\gamma \end{pmatrix} \tag{39}$$

represent the exchange rates with heat bath 1 and 2 respectively. If, as in section 3.1, we are interested in $Q_t$ (the energy i.e. the number of particles exchanged with the left heat bath

at temperature $T_1$ during time $t$), the tilting takes place only on $M_1$ as the current is due to transitions happening to and from the left of the first site. Thus the tilted matrix is

$$M_\lambda = \begin{pmatrix} -(\beta + \gamma) & \alpha e^\lambda + \delta \\ \beta + \gamma e^{-\lambda} & -(\alpha + \delta) \end{pmatrix} \quad . \tag{40}$$

Notice that in the tilted matrix $M_\lambda = M(\mathcal{C}, \mathcal{C}') e^{\lambda q(\mathcal{C}, \mathcal{C}')}$, there is no tilting term on the diagonals because $q(\mathcal{C}, \mathcal{C}) = 0$ and on the off diagonal terms $\lambda$ only shows up for the rates which represent the exchanges at the left boundary.

If instead, we were measuring the flux $\widehat{Q}_t$ between the system and the right heat bath, the titled matrix would be

$$\widehat{M}_\lambda = \begin{pmatrix} -(\beta + \gamma) & \alpha + \delta e^{-\lambda} \\ \beta e^\lambda + \gamma & -(\alpha + \delta) \end{pmatrix} \quad . \tag{41}$$

It is easy to see that $M_\lambda$ and $\widehat{M}_\lambda$ have the same trace and the same determinant. Therefore their largest eigenvalue $\mu(\lambda)$ is the same and for large $t$

$$\langle e^{\lambda Q_t} \rangle \sim \langle e^{\lambda \widehat{Q}_t} \rangle \sim e^{t\mu(\lambda)} \quad .$$

This is rather obvious since particles cannot accumulate in the system and $|Q_t - \widehat{Q}_t| \le 1$.
    One can also notice that the function $\mu(\lambda)$ has the following symmetry

$$\mu(\lambda) = \mu(\lambda') \quad \text{when} \quad e^{\lambda'} = \frac{\gamma \delta}{\alpha \beta} e^{-\lambda} \tag{42}$$

which can again be checked directly as $M_\lambda$ and $M_{\lambda'}$ have the same trace and the same determinant. This is actually a particular case of a general symmetry that is known under the name of the **fluctuation theorem** (see section 4 below). In particular, as we will see below, the symmetry (42) holds for a SSEP of arbitrary size $L$, even though (42) was established so far only for $L = 1$.
    The symmetry (42) is remarkable although the expression of $\mu(\lambda)$ is usually very complicated: already for $L = 1$, one has

$$\mu(\lambda) = \frac{1}{2} \left( \sqrt{(\alpha + \beta + \gamma + \delta)^2 + 4(1 - e^{-\lambda})(\alpha \beta e^\lambda - \gamma \delta)} - (\alpha + \beta + \gamma + \delta) \right) \tag{43}$$

and for $L \ge 1$, $\mu(\lambda)$ is the largest eigenvalue of a $2^L \times 2^L$ matrix $M_\lambda$.

### 3.3   The Legendre transform and the large deviation of the current

The knowledge of the largest eigenvalue $\mu(\lambda)$ of the titled matrix (33) is directly related by the Gartner-Ellis theorem to the large deviation function $I(q)$ of the current by a simple Legendre transform so that for large $t$ one has

$$\mathbb{P}\left(\frac{Q_t}{t} = q\right) \sim e^{-tI(q)} \quad \text{where} \quad I(q) = \sup_{\lambda \in \mathbb{R}} (\lambda q - \mu(\lambda)) \quad . \tag{44}$$

In the example of section 3.2 the symmetry (42) implies that the large deviation function $I(q)$ satisfies

$$I(-q) = I(q) + q \log \frac{\alpha \beta}{\gamma \delta} \quad . \tag{45}$$

This can be easily seen by writing using (42) and (44) that

$$I(q) = \sup_{\lambda \in \mathbb{R}} (\lambda q - \mu(\lambda)) = \sup_{\lambda' \in \mathbb{R}} \left( \left( -\lambda' - \log \frac{\alpha \beta}{\gamma \delta} \right) q - \mu(\lambda') \right) = I(-q) - q \log \frac{\alpha \beta}{\gamma \delta} \quad . \tag{46}$$

We will see in the next section that (45) remains valid for a SSEP of arbitrary length $L$. The equality (45) is another way of formulating the fluctuation theorem. As for $\mu(\lambda)$ this relation is remarkable because even though $I(q)$ is a complicated function of $q$, the difference in the rate functions $I(q) - I(-q)$ is linear in $q$.

The symmetry (45) can be generalized for the ASEP, the asymmetric simple exclusion process. In the ASEP, like for the SSEP, each particle can hop to one of its two neighboring sites on the lattice if the target site is empty, the only difference being that the jumping rates to the left is $r \neq 1$ while the jumping rate to the right is kept at $1$. Then, the symmetry (42) of the large deviation function of the current becomes

$$e^{\lambda'} = r^{L-1} \frac{\gamma\delta}{\alpha\beta} e^{-\lambda} \implies \mu(\lambda) = \mu(\lambda') \tag{47}$$

and (45) is replaced by

$$I(-q) = I(q) + q \left( \log \frac{\alpha\beta}{\gamma\delta} - (L-1)\log r \right) \quad . \tag{48}$$

As for the SSEP, this symmetry is a special case of the fluctuation theorem.

## 4   The Fluctuation Theorem

The Fluctuation Theorem [66, 82, 119] was first discovered for out of equilibrium systems maintained in a steady state by contact with deterministic forces or heat baths [65,71,72,113]. It was then generalized to systems with stochastic heat baths or Markov processes [100, 102, 106] and its derivation is part of the whole field of stochastic thermodynamics [119,120,135].

To derive the Fluctuation Theorem for a discrete time Markov process, consider the special case where the current discussed in section 3.1 is given by $q(\mathcal{C}', \mathcal{C}) = s(\mathcal{C}', \mathcal{C})$ with

$$s(\mathcal{C}_{t+1}, \mathcal{C}_t) = \log \left[ \frac{M(\mathcal{C}_{t+1}, \mathcal{C}_t)}{M(\mathcal{C}_t, \mathcal{C}_{t+1})} \right] \quad . \tag{49}$$

The integrated current $S_t$ over a trajectory $\{\mathcal{C}_0, \cdots \mathcal{C}_t\}$ is

$$S_t = \sum_{\tau=0}^{t-1} s(\mathcal{C}_{\tau+1}, \mathcal{C}_\tau) \quad . \tag{50}$$

As in section 3.1, in the long time limit, the generating function of $S_t$ grows like

$$\langle e^{\lambda S_t} \rangle \sim e^{t\mu(\lambda)} \tag{51}$$

where $e^{\mu(\lambda)}$ is the largest eigenvalue of the tilted matrix whose elements are

$$M_\lambda(\mathcal{C}', \mathcal{C}) = e^{\lambda s(\mathcal{C}', \mathcal{C})} M(\mathcal{C}', \mathcal{C}) = M(\mathcal{C}', \mathcal{C})^{1+\lambda} M(\mathcal{C}, \mathcal{C}')^{-\lambda} \quad . \tag{52}$$

It is then easy to see that $M_\lambda = (M_{-1-\lambda})^T$. This implies the following symmetry, called the Fluctuation Theorem, of the large eigenvalue $\mu(\lambda)$ of the tilted matrix $M_\lambda$

$$\boxed{\mu(\lambda) = \mu(-\lambda - 1)} \quad . \tag{53}$$

At the level of the rate function of $s$, defined as $\mathbb{P}\left(\frac{S_t}{t} = s\right) \asymp e^{-t I(s)}$, the fluctuation theorem takes the following general form (see (44))

$$\boxed{I(s) = I(-s) - s} \tag{54}$$

which follows easily, as in (46), from the fact that $I(s) = \max_\lambda [\lambda s - \mu(\lambda)]$. As we will see below, (45) and (48) are in fact special cases of (54).

## 4.1 Physical interpretation of the Fluctuation Theorem

The integrated current $S_t$ defined in (49,50) is nothing but the change of entropy of the outside world, i.e. here of the heat baths.

Consider first a system in contact with a single heat bath, so that the Markov process satisfies detailed balance (22)

$$M(\mathcal{C}',\mathcal{C})e^{-E(\mathcal{C})/T} = M(\mathcal{C},\mathcal{C}')e^{-E(\mathcal{C}')/T} \quad . \tag{55}$$

Due to the conservation of energy, at each change of configuration $\mathcal{C} \to \mathcal{C}'$, the energy $E_{hb}$ of the heat bath becomes

$$E_{hb} \to E_{hb} + E(\mathcal{C}) - E(\mathcal{C}') \quad . \tag{56}$$

Since the outside world (i.e. the heat bath) is assumed to be big enough, it can absorb energy without modifying its temperature. Therefore its entropy changes by

$$\Delta S_{hb} = \frac{\Delta E_{hb}}{T} = \log\left(\frac{M(\mathcal{C}',\mathcal{C})}{M(\mathcal{C},\mathcal{C}')}\right) \tag{57}$$

where the last equality follows from detailed balance (55). So $S_t$ is simply the change of entropy of the heat bath during time $t$.

This can be generalized to several heat baths at different temperatures, where some jumps satisfy (55) for some temperature $T_1$, others at $T_2$, and so on (see the example of sections 2.5 and 3.2). Then, $S_t$ corresponds to the total increase of entropy of all the heat baths.

The symmetry (54) in the rate function established by the fluctuation theorem implies that the minimum (zero) value of $I(s)$ is at some **positive** value $s^* \geq 0$. Indeed, since $I(s^*) = 0$ we have (see (54))

$$I(-s^*) = s^* \geq 0 \quad . \tag{58}$$

That $s^* \geq 0$ is in agreement with the second law: in a steady state, the most likely situation is that the entropy of the outside world grows with growth rate $s^*$. However, the equality (58) also implies the existence of rare events violating the second law since $I(-s^*) = s^* > 0$ as soon as $s^* \neq 0$.

**Remark:** by a perturbative expansion of $\mu(\lambda)$ around $\lambda = 0$, it is rather easy to show from the definition of $\mu(\lambda)$ as the largest eigenvalue of the tilted matrix (33) and from (49,52) that in the steady state the rate of increase of the entropy of the heat baths is

$$s^* = \left.\frac{d\mu(\lambda)}{d\lambda}\right|_{\lambda=0} = \sum_{\mathcal{C},\mathcal{C}'} M(\mathcal{C},\mathcal{C}')P_{\text{st.}}(\mathcal{C}') \log \frac{M(\mathcal{C},\mathcal{C}')}{M(\mathcal{C}',\mathcal{C})} \tag{59}$$

where $P_{\text{st.}}$ is the steady state measure i.e. is the solution of

$$P_{\text{st.}}(\mathcal{C}) = \sum_{\mathcal{C}'} M(\mathcal{C},\mathcal{C}')P_{\text{st.}}(\mathcal{C}') \qquad \text{with} \quad \sum_{\mathcal{C}} P_{\text{st.}}(\mathcal{C}) = 1 \quad .$$

Using these last relations and (18) one can show that

$$\sum_{\mathcal{C},\mathcal{C}'} M(\mathcal{C},\mathcal{C}')P_{\text{st.}}(\mathcal{C}') \log \frac{P_{\text{st.}}(\mathcal{C}')}{P_{\text{st.}}(\mathcal{C})} = 0$$

so that (59) can be rewritten as

$$s^* = \frac{1}{2}\sum_{\mathcal{C},\mathcal{C}'} \left[M(\mathcal{C},\mathcal{C}')P_{\text{st.}}(\mathcal{C}') - M(\mathcal{C}',\mathcal{C})P_{\text{st.}}(\mathcal{C})\right] \log \frac{M(\mathcal{C},\mathcal{C}')P_{\text{st.}}(\mathcal{C}')}{M(\mathcal{C}',\mathcal{C})P_{\text{st.}}(\mathcal{C})} \quad . \tag{60}$$

Using the fact that $(x-y)\log\frac{x}{y} > 0$ for any pair $x \neq y$ of positive numbers, we see that, in the steady state, the average creation of entropy $s^*$ is strictly positive, as soon as the dynamics does not satisfies detailed balance (22), i.e. as soon as the system is out of equilibrium.

## 4.2   A system in contact with two heat baths

In the example of the SSEP of section 2.5, we have seen (25,26) that the number of particles can be viewed as the energy of a configuration and that the left and right reservoirs as heat baths at temperatures

$$\frac{\alpha}{\gamma} = e^{-1/T_1} \quad ; \quad \frac{\delta}{\beta} = e^{-1/T_2} \quad . \tag{61}$$

This models a system in contact between two heat baths at different temperatures. Using the fluctuation theorem (45), this yields the following identity for the rate function

$$I(-q) = I(q) + q\left(\frac{1}{T_2} - \frac{1}{T_1}\right) \tag{62}$$

and implies that the typical value of the current $q^*$, for which $I(q^*) = 0$, is of the sign of $T_1 - T_2$: typically the energy current $q^*$ flows from high temperature to low temperature.

Relation (62) holds for much more general systems. If $Q_t$ is the energy flowing during time $t$ from a heat bath at temperature $T_1$ to a heat bath at temperature $T_2$, (under mild conditions such that the assumption that the energy of the system considered remains bounded), then the rate function defined as in (44) satisfies (62). When the system jumps from configuration $C_\tau$ to configuration $C_{\tau+1}$, if $q_\tau^{(1)}$ and $q_\tau^{(2)}$ are the energies received by the system from heat baths 1 and 2, the total change of entropy of the heat baths is

$$s_\tau = -\frac{q_\tau^{(1)}}{T_1} - \frac{q_\tau^{(2)}}{T_2} \quad .$$

Then the total integrated current $Q_t$ through the system during time $t$ is

$$Q_t \simeq \sum_{\tau=0}^{t} q_\tau^{(1)} \simeq -\sum_{\tau=0}^{t} q_\tau^{(2)}$$

(where here $A_t \simeq B_t$ means that the difference $|A_t - B_t|$ does not grow with time under the hypothesis that the energy cannot accumulate indefinitely in the system). Then the total change of entropy of the heat baths is

$$S_t \simeq \left(\frac{1}{T_2} - \frac{1}{T_1}\right) Q_t$$

and (62) appears as a special case of (54).

Similarly, for a system in contact with two reservoirs of particles at chemical potentials $\mu_1$ and $\mu_2$, the change of entropy of the reservoirs is (with the convention that the temperature $T = 1$

$$S_t \simeq (\mu_1 - \mu_2) Q_t$$

so that the fluctuation theorem becomes

$$I(-q) = I(q) + q\,(\mu_1 - \mu_2) \quad . \tag{63}$$

## 5 Convexity of the large deviation function of the current

An interpretation of the matrix elements of powers of the tilted matrix (33) is that

$$M_\lambda^t(\mathcal{C}', \mathcal{C}) = \left\langle e^{\lambda Q_t} \middle| \mathcal{C}_t = \mathcal{C}', \mathcal{C}_0 = \mathcal{C} \right\rangle \quad . \tag{64}$$

In the long time limit (for a system with a finite number of configurations $\mathcal{C}$) this is of the form

$$M_\lambda^t(\mathcal{C}', \mathcal{C}) \simeq e^{\mu(\lambda)t} L(\mathcal{C}') R(\mathcal{C})$$

where $L$ and $R$ are the left and right eigenvectors of the matrix $M_\lambda$ associated to the largest eigenvalue $e^{\mu(\lambda)}$ (in the case of a discrete time dynamics)

$$\sum_{\mathcal{C}'} L(\mathcal{C}') M_\lambda(\mathcal{C}', \mathcal{C}) = e^{\mu(\lambda)} L(\mathcal{C}) \quad ; \quad \sum_{\mathcal{C}} M_\lambda(\mathcal{C}', \mathcal{C}) R(\mathcal{C}) = e^{\mu(\lambda)} R(\mathcal{C}')$$

normalized by the relation $\sum_{\mathcal{C}} L(\mathcal{C}) R(\mathcal{C}) = 1$. This leads, for large $t$, to the following distribution of the current

$$\mathrm{Pr}\left( \frac{Q_t}{t} = q \middle| \mathcal{C}_t = \mathcal{C}', \mathcal{C}_0 = \mathcal{C} \right) \simeq \sqrt{\frac{I''(q)\,t}{2\pi}}\, L(\mathcal{C}') R(\mathcal{C})\, e^{-t\,I(q)} \tag{65}$$

where (see(44))

$$I(q) = \max_\lambda \left[ \lambda q - \mu(\lambda) \right] \quad .$$

Given that for any $\tau = \alpha t$ with $0 < \alpha < 1$ and any $\mathcal{C}_\tau$

$$\mathrm{Pr}\left( \frac{Q_t}{t} = \alpha q_1 + (1-\alpha)q_2 \middle| \mathcal{C}_t, \mathcal{C}_0 \right) \geq \mathrm{Pr}\left( \frac{Q_{t-\tau}}{t-\tau} = q_2 \middle| \mathcal{C}_t, \mathcal{C}_\tau \right) \times \mathrm{Pr}\left( \frac{Q_\tau}{\tau} = q_1 \middle| \mathcal{C}_\tau, \mathcal{C}_0 \right) \tag{66}$$

and one can see from (65) that the large deviation function $I(q)$ of the current is convex

$$I\left( \alpha q_1 + (1-\alpha)q_2 \right) \leq \alpha I(q_1) + (1-\alpha)I(q_2) \quad . \tag{67}$$

## 6 Linear response

At equilibrium, the linear response theory relates the fluctuations of a system in absence of driving force to its response to a small driving force.

### 6.1 Einstein relations

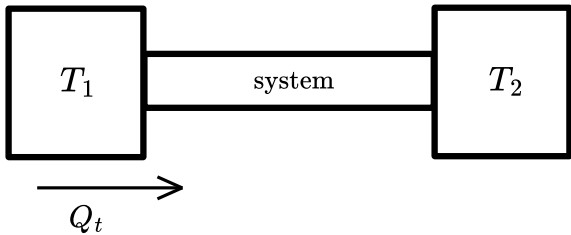

Figure 3: A system in contact with two heat baths. $Q_t$ is the flux of particles from heat bath 1 to heat bath 2 during time $t$.

For a system in contact with two heat baths at temperatures $T_1$ and $T_2$, the current of energy through the system when the difference $T_1 - T_2$ is small is of the form

$$\lim_{t \to \infty} \frac{\langle Q_t \rangle}{t} = \widetilde{D}(T_1) \, (T_1 - T_2) + O\left((T_1 - T_2)^2\right) \tag{68}$$

where $\widetilde{D}$ depends on the system considered. $\widetilde{D}$ is the response coefficient of the system to a small temperature difference between the two heat baths.

On the other hand, at equilibrium, i.e. when $T_1 = T_2$, the average current of energy $\langle Q_t \rangle = 0$ between the two heat baths but there are still fluctuations of energy between them and

$$\text{for } T_1 = T_2, \qquad \lim_{t \to \infty} \frac{\langle Q_t^2 \rangle}{t} = \widetilde{\sigma}(T_1) \quad . \tag{69}$$

The two functions $\widetilde{D}(T)$ and $\widetilde{\sigma}(T)$ depend on the system considered, on its size, on its shape and are complicated functions of temperature. What the linear response theory tells us is that

$$\widetilde{\sigma}(T) = 2 \, k_B \, T^2 \, \widetilde{D}(T) \quad . \tag{70}$$

This Einstein relation is actually a particular case of the fluctuation theorem. Indeed, the total change of entropy of the heat baths during time $t$ is

$$S_t \simeq Q_t \left( \frac{1}{k_B T_2} - \frac{1}{k_B T_1} \right) \quad . \tag{71}$$

Then if for large $t$

$$\langle e^{\alpha Q_t} \rangle \sim e^{t \, \widetilde{\mu}(\alpha)} \tag{72}$$

one should have using (53,71)

$$\widetilde{\mu}(\alpha) = \mu \left( \frac{\alpha}{\frac{1}{k_B T_2} - \frac{1}{k_B T_1}} \right) = \mu \left( -\frac{\alpha}{\frac{1}{k_B T_2} - \frac{1}{k_B T_1}} - 1 \right) = \widetilde{\mu}\left( -\alpha - \frac{1}{k_B T_2} + \frac{1}{k_B T_1} \right) \quad . \tag{73}$$

For $\Delta T = T_1 - T_2$ small this implies that

$$\widetilde{\mu}'(0) = -\widetilde{\mu}'\left( \frac{1}{k_B T_1} - \frac{1}{k_B T_2} \right) = -\widetilde{\mu}'(0) + \frac{\Delta T}{k_B T^2} \widetilde{\mu}''(0) + o(\Delta T) \tag{74}$$

so that

$$\widetilde{\mu}'(0) = \frac{\Delta T}{2 k_B T^2} \widetilde{\mu}''(0) + o(\Delta T)$$

and from (72) one has

$$\lim_{t \to \infty} \frac{\langle Q_t \rangle}{t} = \widetilde{\mu}'(0) \quad \text{and} \quad \lim_{t \to \infty} \frac{\langle Q_t^2 \rangle - \langle Q_t \rangle^2}{t} = \widetilde{\mu}''(0) \tag{75}$$

which leads to (70).

Note that the symmetry (73) of $\widetilde{\mu}$ implies, by a Legendre transform, the symmetry (62).

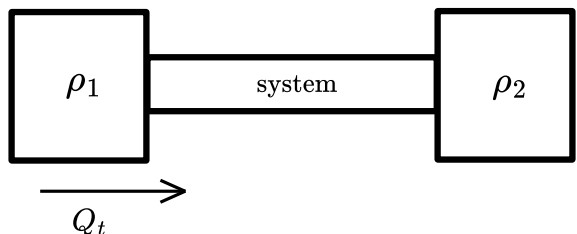

Figure 4: A system in contact with two reservoirs of particles at different densities

## 6.2  Current of particles

There is a version of the Einstein relation (70) , for out of equilibrium systems in contact with two reservoirs of particles at unequal chemical potentials. If, during time $t$, the flux of particles from a reservoir $i$ at chemical potential $\mu_i$ to the system is $Q_t^{(i)}$, the change of entropy of this reservoir $i$ during this time $t$ is

$$\Delta S_i = \frac{\mu_i}{k_B T} Q_t^{(i)} \ .$$

For a system in contact with two reservoirs, let

$$Q_t = -Q_t^{(1)} \simeq Q_t^{(2)}$$

be flux of particles from reservoir **1** to reservoir **2** (we say that $Q_t^{(1)} \simeq -Q_t^{(2)}$ because we assume that the particles cannot accumulate indefinitely in the system). Then the total change of entropy of the reservoirs

$$S_t \simeq \frac{\mu_1 - \mu_2}{T} Q_t \quad .$$

Repeating the same steps as in (71-75), one can show that the generating function of the flux of particles grows exponentially with time

$$\langle e^{\alpha Q_t} \rangle \sim e^{t \tilde{\mu}(\alpha)}$$

and that $\tilde{\mu}$ satisfies

$$\tilde{\mu}(\alpha) = \tilde{\mu}\left(-\alpha - \frac{\mu_1 - \mu_2}{k_B T}\right) \quad .$$

This leads to the the following linear response relation

$$\tilde{\sigma}(\mu_1) = 2 k_B T \, \tilde{D}(\mu_1) \tag{76}$$

where for small $\mu_1 - \mu_2$

$$\lim_{t \to \infty} \frac{\langle Q_t \rangle}{t} = \tilde{D}(\mu_1) \, (\mu_1 - \mu_2) + O\Big((\mu_1 - \mu_2)^2\Big) \tag{77}$$

and at equilibrium (i.e. for $\mu_1 = \mu_2$)

$$\text{For } \mu_1 = \mu_2 \qquad \lim_{t \to \infty} \frac{\langle Q_t^2 \rangle}{t} = \tilde{\sigma}(\mu_1) \quad . \tag{78}$$

   If instead of using chemical potentials, one tries to express the average current and its fluctuations in terms of the densities $\rho_1$ and $\rho_2$ of the two reservoirs

$$\lim_{t \to \infty} \frac{\langle Q_t \rangle}{t} = \hat{D}(\rho_1) \, (\rho_1 - \rho_2) + O\Big((\rho_1 - \rho_2)^2\Big) \tag{79}$$

and

$$\text{For } \rho_1 = \rho_2 \qquad \lim_{t \to \infty} \frac{\langle Q_t^2 \rangle}{t} = \widehat{\sigma}(\rho_1) \tag{80}$$

one has

$$\widehat{\sigma}(\rho_1) = \tilde{\sigma}(\mu_1) \quad \text{and} \quad \widehat{D}(\rho_1) = \tilde{D}(\mu_1) \frac{d\mu_1}{d\rho_1} \quad .$$

Then (76) becomes

$$\widehat{D}(\rho_1) = \frac{\widehat{\sigma}(\rho_1)}{2k_B T} \frac{d\mu_1}{d\rho_1} = \frac{\widehat{\sigma}(\rho_1)}{2k_B T} f''(\rho_1) \tag{81}$$

where $f(\rho)$ is the free-energy per unit volume of a reservoir at density $\rho$. This can be understood by considering a system of $N$ particles in a volume $V$, for which, in the thermodynamic limit, the free energy is extensive, of the form

$$F = V f\left(\frac{N}{V}\right)$$

so that the chemical potential

$$\mu = \frac{dF}{dN} = f'(\rho) \quad \text{and} \quad \frac{d\mu}{d\rho} = f''(\rho) \quad . \tag{82}$$

In the following, we will consider systems of particles in contact with reservoirs at different densities, but at a fixed temperature $T$, and we sill set $k_B T = 1$ so that we will replace the Einstein relation (81) by

$$\frac{2\widehat{D}(\rho)}{\widehat{\sigma}(\rho)} = f''(\rho) \quad . \tag{83}$$

## 6.3 Onsager relations

The fluctuation theorem can be generalized for systems submitted to several small driving forces. For example, for a system evolving according to a discrete time Markov process and in contact with three heat baths at temperatures $T_1, T_2, T_3$, let $q^{(i)}(\mathcal{C}', \mathcal{C})$ be the energy received by the system from the heat bath $i$ when the system jumps from configuration $\mathcal{C}$ to $\mathcal{C}'$ the Markov matrix satisfies

$$M(\mathcal{C}', \mathcal{C}) = M(\mathcal{C}, \mathcal{C}') \exp\left[ -\frac{q^{(1)}(\mathcal{C}', \mathcal{C})}{k_B T_1} - \frac{q^{(2)}(\mathcal{C}', \mathcal{C})}{k_B T_2} - \frac{q^{(3)}(\mathcal{C}', \mathcal{C})}{k_B T_3} \right] \tag{84}$$

which expresses the fact that each heat bath tries to equilibrate the system at its own temperature.

Given an initial configuration $\mathcal{C}_0$ and a final configuration $\mathcal{C}_t$, if $Q_t^{(i)}$ is the total energy received that from heat bath $i$ during $t$ time steps,

$$\text{Pro}(\{Q_t^{(i)}\}|\mathcal{C}_0, \mathcal{C}_t) = \sum_{\mathcal{C}_1} \cdots \sum_{\mathcal{C}_{t-1}} \left[ \prod_{\tau=0}^{t-1} M(\mathcal{C}_{\tau+1}, \mathcal{C}_\tau) \right] \times \prod_{i=1}^{3} \delta\left( Q_t^{(i)} - \sum_{\tau=0}^{t-1} q^{(i)}(\mathcal{C}_{\tau+1}, \mathcal{C}_\tau) \right) \quad .$$

Using the detailed balance condition (84) one gets

$$\text{Pro}\left(\{Q_t^{(i)}\}|\mathcal{C}_0, \mathcal{C}_t\right) = \text{Pro}\left(-\{Q_t^{(i)}\}|\mathcal{C}_t, \mathcal{C}_0\right) \exp\left[ -\sum_i \frac{Q^{(i)}(\mathcal{C}', \mathcal{C})}{k_B T_i} \right] \quad . \tag{85}$$

In the long time limit, if one assumes that the energy cannot accumulate in the system i.e. $Q_t^{(3)} \simeq -Q_t^{(1)} - Q_t^{(2)}$ and that up to non-exponential factors, $\text{Pro}(\{Q_t^{(i)}\}|\mathcal{C}_0, \mathcal{C}_t)$ does not depend on the initial and the final configurations $\mathcal{C}_0, \mathcal{C}_t$, one has for large $t$

$$\text{Pro}\left( \frac{Q_t^{(1)}}{t} = q_1, \frac{Q_t^{(2)}}{t} = q_2 \right) \sim e^{-tI(q_1,q_2)}$$

then (85) implies that

$$I(q_1, q_2) = I(-q_1, -q_2) - q_1\left( \frac{1}{k_B T_3} - \frac{1}{k_B T_1} \right) - q_2\left( \frac{1}{k_B T_3} - \frac{1}{k_B T_2} \right)$$

which generalizes (62) to the case of three heat baths.

Therefore if in the long time limit

$$\left\langle \exp\left( \alpha_1 Q^{(1)}(t) + \alpha_2 Q^{(2)}(t) \right) \right\rangle \sim e^{t\, \widetilde{\mu}(\alpha_1, \alpha_2)}$$

with

$$\widetilde{\mu}(\alpha_1, \alpha_2) = \widetilde{\mu}\left( -\alpha_1 - \frac{1}{k_B T_3} + \frac{1}{k_B T_1}, -\alpha_2 - \frac{1}{k_B T_3} + \frac{1}{k_B T_2} \right) \ .$$

Then proceeding as in (74,75) when $\Delta T_1 = T_1 - T_3$ and $\Delta T_2 = T_2 - T_3$ are small, one gets at linear order in the perturbation $\Delta T_1$ and $\Delta T_2$

$$\lim_{t \to \infty} \begin{bmatrix} \frac{\langle Q_1 \rangle}{t} \\ \frac{\langle Q_2 \rangle}{t} \end{bmatrix} = \begin{bmatrix} \mathcal{M}_{11} & \mathcal{M}_{12} \\ \mathcal{M}_{21} & \mathcal{M}_{22} \end{bmatrix} \begin{bmatrix} \Delta T_1 \\ \Delta T_2 \end{bmatrix} \tag{86}$$

where the response matrix

$$\mathcal{M}_{i,j} = \frac{1}{2k_B T^2} \frac{\partial^2 \widetilde{\mu}(0,0)}{\partial \alpha_i\, \partial \alpha_j} \ .$$

We see that the matrix $\mathcal{M}_{ij}$ is symmetric

$$\boxed{\mathcal{M}_{12} = \mathcal{M}_{21}} \ . \tag{87}$$

This an example of the Onsager reciprocity relation.

# 7    Diffusive systems

By diffusive systems, one means that, for a one dimensional system of length $L$, the transport coefficients $\widehat{D}, \widehat{\sigma}$ scale, for large $L$, as $1/L$.

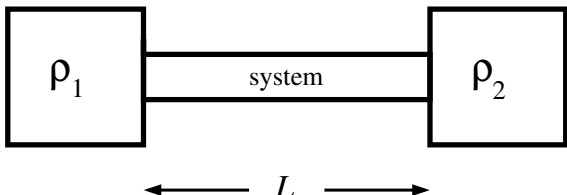

Figure 5: In diffusive systems, the transport coefficients $\widehat{\sigma}$ and $\widehat{D}$ decay like the inverse of the system size $L$

## 7.1 Fick's law

In the case of transport of particles this is called **Fick's law** with the following large $L$ behavior of the transport coefficients defined in section 6.2

$$\widehat{\sigma}(\rho) \simeq \frac{\sigma(\rho)}{L} \quad ; \quad \widehat{D}(\rho) \simeq \frac{D(\rho)}{L} \quad .$$

This implies that the steady state current of particles in one dimension is of the form

$$
\begin{aligned}
\frac{\langle Q_t \rangle}{t} &= \frac{D(\rho_1)}{L}(\rho_1 - \rho_2) && \text{for} \quad \rho_1 - \rho_2 \ \text{small} \\
\frac{\langle Q_t^2 \rangle}{t} &= \frac{\sigma(\rho_1)}{L} && \text{for} \quad \rho_1 = \rho_2 \quad .
\end{aligned}
\tag{88}
$$

In higher dimension, this becomes in particular $J = -D(\rho)\nabla\rho$.

## 7.2 Fourier's law

Similarly, in the case of transport of energy, if the large $L$ behavior of the transport coefficients defined in (68,69) is of the form

$$\widehat{\sigma}(T) \simeq \frac{\sigma(T)}{L} \quad ; \quad \widehat{D}(T) \simeq \frac{D(T)}{L}$$

one says that the system satisfies Fourier's law with a steady state current of energy in one dimension of the form

$$\frac{\langle Q_t \rangle}{t} = \frac{D(T_1)}{L}(T_1 - T_2) \quad \text{for} \quad T_1 - T_2 \ \text{small}$$

which becomes in higher dimension $J = -D(T)\nabla T$.

**Remark:** not all systems satisfy Fick's or Fourier's law: the simplest examples is the cases of an ideal gas and of the harmonic chain [112] for which $\widehat{D}$ and $\widehat{\sigma}$ do not decay with $L$. Also a number of mechanical interacting systems with non linear forces often exhibit an anomalous Fourier's law [61, 103, 128], where $\widehat{D}$ and $\widehat{\sigma}$ decay as non integer powers of the system size $L$.

# 8 How to determine the transport coefficients

The transport coefficients $D(\rho)$ and $\sigma(\rho)$ are macroscopic properties of a given system. They are related (see (83)) by

$$\boxed{2D(\rho) = \sigma(\rho)\, f''(\rho)} \quad . \tag{89}$$

If the free energy $f(\rho)$ is known, the question then is to determine one of them starting from the definition of a microscopic model.

## 8.1 The SSEP

At equilibrium, i.e. if $\alpha\beta = \gamma\delta$, for the dynamics to satisfy detailed balance (see section 2.5) the energy of a configuration $E_N$ must depend only on the number of particles $N$ in the configuration and it should be linear in $N$, i.e. $E_N = Ne_0$. with $\beta e_0 = \log(\gamma/\alpha)$. The partition function is therefore

$$Z_L(N) = \frac{L!}{N!(L-N)!} e^{-N\beta e_0}$$

and, for large $L$ (setting $k_B T = 1$), the free energy $f(\rho)$ per unit volume is

$$f(\rho) = \rho e_0 + \rho \log \rho + (1-\rho) \log(1-\rho) \quad \text{so that} \quad f''(\rho) = \frac{1}{\rho(1-\rho)} \tag{90}$$

and from (89)

$$\sigma(\rho) = 2\,\rho\,(1-\rho)\,D(\rho) \quad . \tag{91}$$

Let us now determine the steady state current. If $n_i = 0$ or $1$ is the occupation of site $i$, its evolution is given by

$$\frac{d\langle n_i \rangle}{dt} = \langle n_{i-1}(1-n_i) \rangle + \langle n_{i+1}(1-n_i) \rangle - \langle n_i(1-n_{i+1}) \rangle - \langle n_i(1-n_{i-1}) \rangle$$

$$= \langle n_{i-1} \rangle + \langle n_{i+1} \rangle - 2\langle n_i \rangle \tag{92}$$

and at the boundaries

$$\frac{d\langle n_1 \rangle}{dt} = \alpha + \langle n_2 \rangle - (1 + \alpha + \gamma)\langle n_1 \rangle$$

$$\frac{d\langle n_L \rangle}{dt} = \delta + \langle n_{L-1} \rangle - (1 + \beta + \delta)\langle n_L \rangle \quad . \tag{93}$$

This leads to the following steady state profile

$$\langle n_i \rangle = \frac{\rho_1(L - i + b) + \rho_2(i + a - 1)}{L + a + b - 1} \tag{94}$$

where

$$a = \frac{1}{\alpha + \gamma} \quad ; \quad b = \frac{1}{\beta + \delta} \quad ; \quad \rho_1 = \alpha\,a \quad ; \quad \rho_2 = \delta\,b \quad ; \tag{95}$$

and the steady state current (between site $i$ and site $i+1$) is given by

$$J_i = \langle n_i(1 - n_{i+1}) - n_{i+1}(1 - n_i) \rangle = \langle n_i - n_{i+1} \rangle = \frac{\rho_1 - \rho_2}{L + a + b - 1} \quad . \tag{96}$$

We see that for large $L$, the current decays like $1/L$ for large $L$ in agreement with Fick's law and that the transport coefficient $D(\rho) = 1$ (see section 7.1) so that for the SSEP

$$\boxed{D(\rho) = 1 \quad ; \quad \sigma(\rho) = 2\rho(1-\rho)} \quad . \tag{97}$$

**Remark 1:** the steady state current (96) does not depend on $i$ simply because particles cannot accumulate.

**Remark 2:** the parameters $a$ and $b$ in (94-96) are related to the strength of the contacts with the two reservoirs. For example a large value of $a$ corresponds to a weak contact with the left reservoir making the steady state resemble the equilibrium at density $\rho_2$. This illustrates the fact that the steady state depends on the strengths of the contacts with the reservoirs [32, 33, 49, 70, 77].

**Remark 3:** for the SSEP, one can write evolution equations which generalize (92,93) for higher correlation functions. For example for the two point function one has

$$\frac{d\langle n_i n_j \rangle}{dt} = \begin{cases} \langle(n_{i+1} + n_{i-1} - 2n_i)n_j\rangle + \langle(n_{j+1} + n_{j-1} - 2n_j)n_i\rangle & i, j \text{ not neighbors} \\ \langle n_{i+1}n_{i-1}\rangle + \langle n_i n_{i+2}\rangle - 2\langle n_i n_{i+1}\rangle & j = i + 1 \end{cases} \tag{98}$$

plus additional equations which generalize (93) to take into account the boundary effects. These equations can be solved in the steady state. One then finds [56, 126] for $1 \leq i < j \leq L$:

$$\langle n_i n_j \rangle_c \equiv \langle n_i n_j \rangle - \langle n_i \rangle \langle n_j \rangle = -(\rho_1 - \rho_2)^2 \frac{(i + a - 1)(L + b - j)}{(L + a + b - 1)^2 (L + a + b - 2)} \quad . \tag{99}$$

Similarly for the 3-point function one gets for $1 \leq i < j < k \leq L$:

$$\langle n_i n_j n_k \rangle_c = -2(\rho_1 - \rho_2)^3 \frac{(i + a - 1)(L + 1 + b - a - 2j)(L + b - k)}{(L + a + b - 1)^3 (L + a + b - 2)^2 (L + a + b - 3)} \tag{100}$$

(where the truncated correlation function is defined by
$\langle n_i n_j n_k \rangle_c = \langle n_i n_j n_k \rangle - \langle n_i n_j \rangle \langle n_k \rangle - \langle n_i n_k \rangle \langle n_j \rangle - \langle n_j n_k \rangle \langle n_i \rangle + 2 \langle n_i \rangle \langle n_j \rangle \langle n_k \rangle$).

## 8.2 Gradient models

The SSEP is an example of a gradient model. In gradient models, the steady state current between two neighboring $i$ and $i + 1$ sites can be written as a difference of the form

$$J_i = \left\langle \phi(n_{i-k-1}, n_{i-k}, \cdots n_{i+\ell}) \right\rangle - \left\langle \phi(n_{i-k}, n_{i-k+1} \cdots, n_{i+\ell+1}) \right\rangle \tag{101}$$

(see (96)). Adding over all sites $i$, one gets a telescopic sum so that

$$\frac{1}{L} \sum_{j=1}^{L} J_i \simeq \frac{1}{L} \left[ \left\langle \phi(n_{i-k-1}, n_{i-k}, \cdots n_{i+\ell}) \right\rangle_{\text{equ. at } \rho_1} - \left\langle \phi(n_{i-k-1}, n_{i-k}, \cdots n_{i+\ell}) \right\rangle_{\text{equ. at } \rho_2} \right] \tag{102}$$

and this gives

$$D(\rho) = \frac{d}{d\rho} \left\langle \phi(n_{i-k-1}, n_{i-k}, \cdots n_{i+\ell}) \right\rangle_{\text{equ. at } \rho} \quad . \tag{103}$$

As an example consider an exclusion process model with the following jump rates (see Figure 6). A particle jumps to each of its immediate neighbors with rate 1 if the target site is empty. Otherwise it may jump to its second neighbor with rate $\alpha$.

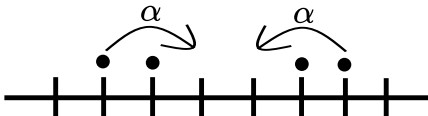

Figure 6: a particle can jump to each neighbor at rate 1 and if this neighbor is occupied, it jumps to the second neighbor at rate $\alpha$

In this case we obtain for the steady state particle current $J_i$ through the bond $i, i + 1$ is

$$J_i = \alpha \langle n_{i-1} n_i (1 - n_{i+1}) \rangle + 1 \langle n_i (1 - n_{i+1}) \rangle + \alpha \langle n_i n_{i+1} (1 - n_{i+2}) \rangle - $$
$$- \alpha \langle n_{i+2} n_{i+1} (1 - n_i) \rangle - 1 \langle n_{i+1} (1 - n_1) \rangle - \alpha \langle n_{i+1} n_i (1 - n_{i-1}) \rangle \tag{104}$$

i.e.

$$J_i = \langle n_i \rangle - \langle n_{i+1} \rangle + \alpha \big( \langle n_{i-1} n_i \rangle + \langle n_i n_{i+1} \rangle - \langle n_i n_{i+1} \rangle - \langle n_{i+1} n_{i+2} \rangle \big) \quad . \tag{105}$$

Proceeding further like before we get

$$D = 1 + 4\rho\alpha \tag{106}$$

where we have used the fact that at equilibrium, detailed balance is satisfied when all configurations are equally likely (implying that $< n_i n_{i+1} > = \rho^2$ with $(\rho)$ is given by (90). Therefore

$$\sigma(\rho) = 2\rho(1-\rho)\,D(\rho) \quad . \tag{107}$$

Other diffusive systems for which the transport coefficients are known include

- The KMP model introduced by Kipnis Marchioro Presutti [96]

  In this model there is an energy $\rho_i$ on each site and during every infinitesimal time interval $dt$, the energies of each pair of neighboring sites have a probability $2dt$ of being updated in the following way

  $$\left(\rho_i, \rho_{i+1}\right) \quad \rightarrow \quad \left((\rho_i + \rho_{i+1})z\,,\,(\rho_i + \rho_{i+1})(1-z)\right)$$

  where $z$ is a random number distributed uniformly between in the interval $(0,1)$. For this model it is easy to check that the partition function $Z_L(N) = N^{L-1}/(L-1)!$ for a system for an isolated system with $\sum_i \rho_i = N$, and that $D(\rho) = 1$ so that $f(\rho) = 1 - \log(\rho)$ and (see (89))

  $$D(\rho) = 1 \quad ; \quad \sigma(\rho) = 2\rho^2 \quad . \tag{108}$$

- The zero range process

  In the zero range process [69], there is on each site an integer number $n_i$ of particles. During every infinitesimal time interval $dt$, a particle jumps from site $i$ to site $i+1$ with rate $u(n_i)$ and similarly a particle jumps from site $i$ to site $i-1$ with i the same rate $u(n_i)$. The jump rates $u(n)$ are in principle arbitrary. Clearly the zero range model is gradient since

  $$J_i = \langle u(n_i)\rangle - \langle u(n_{i+1})\rangle \quad . \tag{109}$$

  Then one can show (see Appendix A) that at equilibrium

  $$\langle u(n)\rangle = e^{f'(\rho)}$$

  where $f(\rho)$ is the free energy per unit volume at density $\rho$ (which depends on all the $u(n)$'s which define the model). As a consequence (see Appendix A)

  $$D(\rho) = \frac{\sigma'(\rho)}{2} = f''(\rho)\,e^{f'(\rho)} \quad . \tag{110}$$

- Non-interacting particles

  A special case of the zero range process is when $u(n) = n$ which corresponds to non-interacting particles. Then $f(\rho) = \rho \log \rho - \rho$ and

  $$D(\rho) = 1 \quad ; \quad \sigma(\rho) = 2\rho \quad . \tag{111}$$

**Remark:** for non gradient models, i.e. when the current can not be written as a gradient [94] as in (101), there is usually not an explicit expression of $D(\rho)$ but there exist some approximations [5] based on an exact variational principle [127]. One way to explain the origin of this variational principle is to remember that $D(\rho)$ and $\sigma(\rho)$ are derivatives of the largest eigenvalue of a tilted matrix (see(68,69,75)) and that finding the largest eigenvalue of a matrix can also be formulated as a variational principle.

# 9 Fluctuating Hydrodynamics

Fluctuation hydrodynamics gives a way of describing the evolution and the fluctuations of a system at a macroscopic scale.

Consider a one dimensional lattice gas of $L$ sites for which each configuration is characterized by the number $n_i$ of particles on site $i$. When $L$ is very large, one can introduce a continuous macroscopic position $x$ defined by

$$i = Lx \qquad \text{where} \quad 0 < x < 1 \quad . \tag{112}$$

Because we consider here diffusive systems, one can also define a macroscopic time $\tau$ related to the time $t$ of the microscopic model by

$$t = L^2 \tau \quad . \tag{113}$$

Now if we divide a very large system $L$ into large boxes of size $l$ as in figure 7, one can define

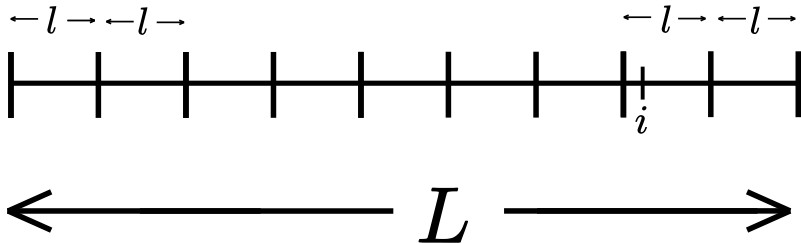

Figure 7: One can divide a system of size $L$ into large boxes of size $l$ with $1 \ll l \ll L$

at the macroscopic level a density $\rho(x, \tau)$ as

$$\rho(x, \tau) = \frac{1}{l} \sum_{i=Lx}^{Lx+l-1} n_i(t)$$

which is the density of particles in the box of size $l$ starting at $Lx$. One can also associate to this density $\rho(x, \tau)$ a macroscopic current $j(x, \tau)$ and the conservation of the number of particles implies that

$$\boxed{\partial_\tau \rho = -\partial_x j} \quad . \tag{114}$$

Fluctuating hydrodynamics [127] tells us how, for a system of large size $L$, this macroscopic current fluctuates

$$\boxed{j(x, \tau) = -D(\rho(x, \tau))\rho'(x, \tau) + \sqrt{\frac{\sigma(\rho(x, \tau))}{L}}\, \eta(x, \tau)} \tag{115}$$

where $\eta(x, \tau)$ is a Gaussian white noise which satisfies

$$\langle \eta(x, \tau) \rangle = 0 \quad ; \quad \langle \eta(x, \tau)\eta(x', \tau') \rangle = \delta(x - x')\delta(\tau - \tau') \quad . \tag{116}$$

This way, (114) and (115) give the stochastic evolution the macroscopic density profile $\rho(x, \tau)$ as a function of the macroscopic time $\tau$.

**Remark 1:** in (115) we see that the average of the current over the noise $\eta(x, \tau)$ is

$\langle j(x,\tau)\rangle = -D(\rho(x,\tau))\rho'(x,\tau)$ which is Fick's law (88).

**Remark 2:** one way to relate the microscopic current of a given diffusive model to the macroscopic current $j(x,\tau)$ is to consider the time integrated current $Q_i(t)$ (of the microscopic model) between site $i$ and site $i+1$ and to write that

$$Q_i(t) \simeq L \int_0^\tau j(x,\tau')d\tau' \tag{117}$$

(with $i = Lx$ and $t = L^2\tau$ as in (112,113)).

To check that the factor $L$ in (117) consider a one dimensional system at equilibrium at density $\rho$, i.e. such that $\rho(x,\tau) = \rho$ and let $q_\tau$ be the average over the system of the macroscopic integrated current

$$q_\tau = \int_0^1 dx \int_0^\tau j(x,\tau')d\tau' \quad .$$

From (115) one gets that

$$\langle q_\tau^2 \rangle = \frac{\sigma(\rho)}{L}\tau$$

in agreement with Fick's law (88) since $\langle Q_i(t)^2 \rangle = L^2\langle q_\tau^2 \rangle = L\sigma(\rho)\tau = \frac{\sigma(\rho)}{L}t$ (see (117,113)).

# 10 Steady state profile and pair long range correlations in diffusive systems

## 10.1 General transport coefficients

For a chain of length $L$ in contact with two reservoirs at densities $\rho_1$ on the left and $\rho_2$ on the right, the evolution (114,115) becomes deterministic in the large $L$ limit

$$\partial_\tau\rho = \partial_x\big(D(\rho)\partial_x\rho\big)$$

and the steady state current $j^*$ and profile $\rho^*(x)$ are given by

$$\boxed{j^* = -D(\rho^*)\partial_x\rho^* = \int_{\rho_2}^{\rho_1} D(\rho)d\rho \quad ; \quad x\,j^* = \int_{\rho^*(x)}^{\rho_1} D(\rho)d\rho} \quad . \tag{118}$$

For large $L$, the fluctuations of the density around the steady state profile $\rho^*$ are small

$$\rho(x,\tau) = \rho^*(x) + \frac{1}{\sqrt{L}}r(x,\tau) + o\left(\frac{1}{\sqrt{L}}\right) \tag{119}$$

and according to (114,115)

$$\partial_\tau r(x,\tau) = \partial_{xx}^2\big(D(\rho^*(x))r(x,\tau)\big) - \partial_x\big(\sqrt{\sigma(\rho^*(x))}\,\eta(x,\tau)\big) \quad . \tag{120}$$

One can then compute the two-point correlations in the steady state [115, 126, 127]. Using the Green function $G(x,\tau|y)$ defined as the solution of

$$\partial_\tau G(x,\tau|y) = \partial_{xx}^2\big(D(\rho^*(x))G(x,\tau|y)\big) = D(\rho^*(y))\partial_{yy}^2 G(x,\tau|y) \tag{121}$$

with $G(0, \tau | y) = G(1, \tau | y) = 0$ and $G(x, 0 | y) = \delta(x - y)$, the solution of (120) can be written as

$$r(x, \tau) = \int_{-\infty}^{\tau} d\tau' \int_0^1 dy \, G(x, \tau - \tau' | y) \left[ -\partial_y \left( \sqrt{\sigma(\rho^*(y))} \, \eta(y, \tau') \right) \right]$$

which can be rewritten (after an integration by part) as

$$r(x, \tau) = \int_{-\infty}^{\tau} d\tau' \int_0^1 dy \left( \frac{dG(x, \tau - \tau' | y)}{dy} \right) \sqrt{\sigma(\rho^*(y))} \, \eta(y, \tau') \quad .$$

Then by averaging over the white noise $\eta$ one gets

$$\langle r(x, \tau) r(y, \tau) \rangle = \int_0^{\infty} ds \int_0^1 dz \left( \frac{dG(x, s | z)}{dz} \right) \left( \frac{dG(y, s | z)}{dz} \right) \sigma(\rho^*(z)) \quad .$$

Doing a few more integrations by part as in equations (17-23) of [115] and using the properties of the Green function one gets

$$\langle r(x, \tau) r(y, \tau) \rangle = \frac{\sigma(\rho^*(x))}{2D(\rho^*(x))} \delta(x - y) + C(x, y)$$

where

$$C(x, y) = -j^* \int_0^1 dz \, \frac{d}{dz} \left( \frac{\sigma'(\rho^*(z))}{2D(\rho^*(z))} \right) \left( \int_0^{\infty} ds \, G(x, s | z) G(y, s | z) \right) \tag{122}$$

which gives for the equal time correlations through (119) and (89)

$$\boxed{ \langle \rho(x) \rho(y) \rangle_c = \frac{\delta(x - y)}{L \, f''(\rho^*(x))} + \frac{C(x, y)}{L} } \tag{123}$$

where the first term represents the local part and the second term represents the long range correlations.

**Remark 1:** in (122) one can see that in general, at equilibrium (i.e. because at equilibrium $j^* = 0$), the long range part of the correlations vanishes. It also vanishes out of equilibrium for some particular models like the zero range process (110) or the non-interacting particles (111) $\left( \text{because for these models } \sigma'(\rho) = 2D(\rho) \right)$.

**Remark 2:** physically the origin of the correlations comes from a response of the densities at positions $x$ and $y$ to a fluctuation of the current in the past (see figure 8).

## 10.2 The case of the SSEP

For the SSEP, $D(\rho) = 1$ and $\sigma(\rho) = 2\rho(1 - \rho)$ (see (97)). Thus the Green function is given by

$$G(x, \tau | y) = 2 \sum_{n \geq 1} \sin(n \pi x) \sin(n \pi y) \, e^{-n^2 \pi^2 \tau} \quad .$$

This gives for $x < y$

$$\int_0^1 dz \int_0^{\infty} ds \, G(x, s | z) G(y, s | z) = \frac{x(1 - y)}{2} \quad .$$

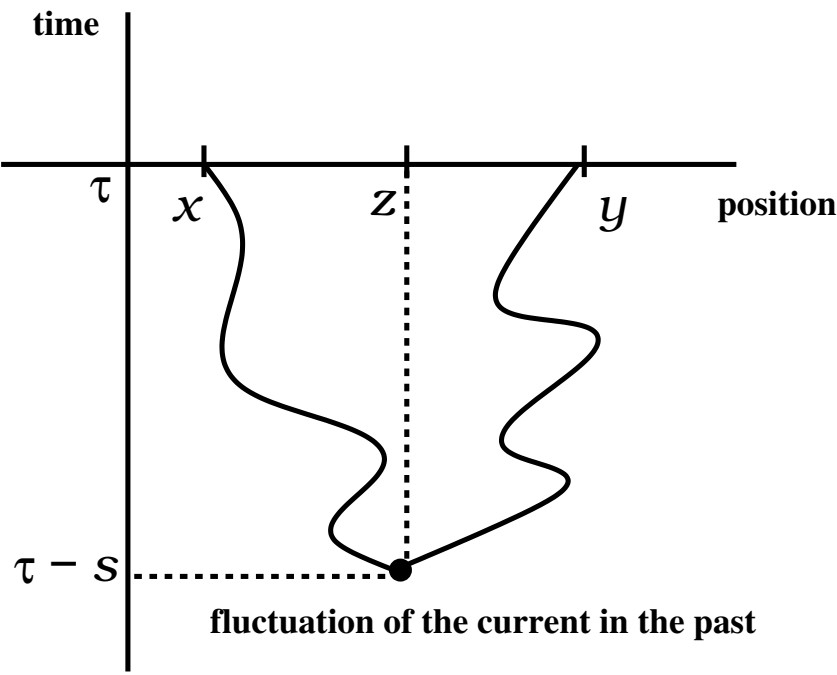

Figure 8: at time $\tau$ the correlations are due to the response of the densities at different points $x$ and $y$ to the same fluctuation of the current at position $z$ and time $\tau - s$ in the past.

As (see (118,122))

$$j^* = \rho_1 - \rho_2 \quad ; \quad \rho^*(x) = (1-x)\rho_1 + x\rho_2 \tag{124}$$

one has from (123)

$$\left\langle \rho(x)\rho(y) \right\rangle_c = \frac{\rho^*(x)\left(1-\rho^*(x)\right)\delta(x-y)}{L} - \frac{(\rho_1-\rho_2)^2}{L}\left[x(1-y)\mathbb{1}_{y>x} + y(1-x)\mathbb{1}_{x>y}\right] \tag{125}$$

On the other hand, if the parameters $a$ and $b$ in (95) are fixed (i.e. do not grow with system size $L$) the expressions (94,96,99) of the microscopic calculation become for large $L$, writing $i = Lx < j = Ly$

$$\langle n_i \rangle = \rho^*(x) \quad ; \quad \langle n_i^2 \rangle - \langle n_i \rangle^2 = \rho^*(x)\left(1-\rho^*(x)\right) \tag{126}$$

$$\langle n_i n_j \rangle - \langle n_i \rangle \langle n_j \rangle = -\frac{(\rho_1-\rho_2)^2}{L}x(1-y) \quad . \tag{127}$$

We see that the results of the microscopic calculations (126,127) and of the fluctuating hydrodynamics calculation (124,125) fully agree.

**Remark:** the long range part of the correlations (123,127) is of order $1/L$. Naively one could believe that these correlations do not matter in the large $L$ limit. This conclusion is not totally true: for example consider of the number of particles $N = \sum_{i=1}^{L} n_i$. One has in the steady state

$$\langle N^2 \rangle - \langle N \rangle^2 = \sum_{i=1}^{L}\left[\langle n_i^2 \rangle - \langle n_i \rangle^2\right] + 2\sum_{i<j}\left[\langle n_i n_j \rangle - \langle n_i \rangle \langle n_j \rangle\right] \tag{128}$$

While the summand of the second summ is of order $\frac{1}{L}$, there are $O(L^2)$ terms in the sum : hence, the second sum is of the same order as the first one. This gives in the large $L$ limit

$$\langle N^2 \rangle - \langle N \rangle^2 = L\left( \frac{\rho_1^2 - \rho_2^2}{2(\rho_1 - \rho_2)} - \frac{1}{3}\frac{\rho_1^3 - \rho_2^3}{\rho_1 - \rho_2} - \frac{1}{12}(\rho_1 - \rho_2)^2 \right) \qquad (129)$$

where the last term represents the contribution of the sum over the long range correlations.

## 11   Large deviations from the macroscopic fluctuation theory

We are going now to discuss the large deviation function of the density profile [93, 95]. We want to estimate the probability of observing a certain density profile $\rho(x, \tau)$. Physically the main idea is that although the profile $\rho$ might be very far from the steady state profile $\rho^*$ as in figure 9, locally the difference of density between two nearby macroscopic positions $x$ and $x + \Delta x$ will be small, and if $\Delta x$ is such that $\Delta x \ll 1 \ll L\Delta x$, the macroscopic interval $(x, x + \Delta x)$ contains lots of sites. Therefore, locally the interval $(x, x + \Delta x)$ is close to equilibrium, and one can use fluctuating hydrodynamics at the level of this interval. So we can

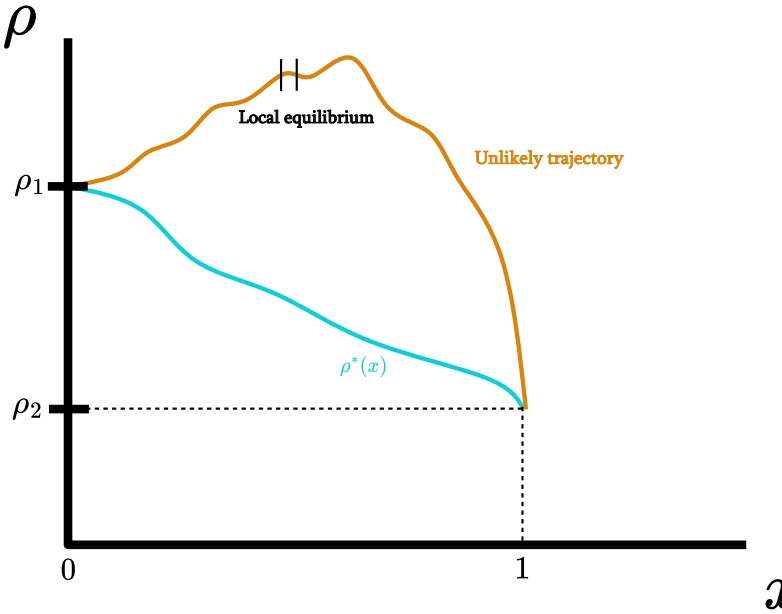

Figure 9: For a macroscopic profile $\rho$, the system is locally close to equilibrium even if the global profile is far from the steady state profile $\rho^*$. Physically this is why one can use locally fluctuating hydrodynamics.

write from (115), the probability of observing a density and current profile $\{\rho(x, \tau'), j(x, \tau')\}$ during the time $\tau_0 \leq \tau' \leq \tau$ given the initial density profile $\rho(x, \tau_0)$

$$\mathbb{P}\big(\{\rho(x, \tau'), j(x, \tau')\}\big) \propto \exp\left( -L \int_0^1 dx \int_{\tau_0}^{\tau} d\tau' \frac{(j + D(\rho)\rho')^2}{2\sigma(\rho)} \right) \qquad (130)$$

which is simply a way of rewriting the Gaussian distribution of the white noise $\eta$ in (115). Therefore the probability of observing a profile $\rho(x, \tau)$ given an initial profile $\rho(x, \tau_0)$ is

$$\mathbb{P}\big(\rho(x, \tau)|\rho(x, \tau_0)\big) \propto \int \mathcal{D}j\mathcal{D}\rho \exp\left( -L \int_0^1 dx \int_{\tau_0}^{\tau} d\tau' \frac{(j + D(\rho)\rho')^2}{2\sigma(\rho)} \right) \qquad (131)$$

where the (path) integral is over all currents $j(x, \tau')$ and density profiles $\rho(x, \tau')$ that satisfy the conservation equation $\partial_{\tau'}\rho = -\partial_x j$, the initial condition $\rho(x, \tau' = 0) = \rho(x, \tau_0)$ and the boundary conditions $\rho(0, \tau) = \rho_1$, $\rho(1, \tau) = \rho_2$.

The **macroscopic fluctuation theory** [12, 13, 19] is based on the idea that, for large $L$, this path integral is dominated by one (or possibly several) saddle trajectories $\{j(x, \tau'), \rho(x, \tau')\}$ for $\tau_0 \le \tau \le \tau'$.

## 11.1 The Hamilton-Jacobi equation

For large $L$, let $\mathcal{F}_\tau(\rho(x, \tau))$ be the large deviation function of the density, given an initial profile $\rho(x, 0)$. This means that

$$\mathbb{P}\big(\rho(x, \tau)|\rho(x, 0)\big) \propto \exp\left[-L\,\mathcal{F}_\tau(\rho(x, \tau))\right]$$

where the functional $\mathcal{F}_\tau$ is given by

$$\mathcal{F}_\tau\big(\rho(x, \tau)\big) = \min_{\{\rho(x,\tau'), j(x,\tau')\}} \left[\int_0^1 dx \int_{\tau_0}^\tau d\tau' \frac{(j + D(\rho)\rho')^2}{2\sigma(\rho)}\right] . \tag{132}$$

If the optimal profile is such that for $\delta\tau$ small, $q(x) = j(x, \tau)$ and $\rho(x, \tau - \delta\tau) = \rho(x, \tau) - \delta\rho$, one should have

$$\mathcal{F}_\tau\big(\rho(x)\big) = \min_{q(x), \delta\rho(x)} \left[\mathcal{F}_{\tau-\delta\tau}(\rho(x) - \delta\rho(x)) + \delta\tau \int_0^1 dx \frac{\big(q(x) + D(\rho(x))\rho'(x)\big)^2}{2\sigma(\rho(x))}\right] \tag{133}$$

which gives

$$\frac{d\mathcal{F}_\tau}{d\tau} = \min_{q(x), \delta\rho(x)} \int_0^1 dx \left[-\frac{\delta\mathcal{F}_\tau}{\delta\rho(x)}\frac{\delta\rho(x)}{\delta\tau} + \frac{\big(q(x) + D(\rho(x))\rho'(x)\big)^2}{2\sigma(\rho(x))}\right] .$$

Then due to the conservation relation (114) one has $\frac{\delta\rho(x)}{\delta\tau} = -\frac{dq(x)}{dx}$ and one gets after an integration by part

$$\frac{d\mathcal{F}_\tau}{d\tau} = \min_{q(x), \delta\rho(x)} \int_0^1 dx \left[-q(x)\frac{d}{dx}\left(\frac{\delta\mathcal{F}_\tau}{\delta\rho(x)}\right) + \frac{\big(q(x) + D(\rho(x))\rho'(x)\big)^2}{2\sigma(\rho(x))}\right] .$$

Under this form it is easy to optimize over $q(x)$ and one finds that the optimal current at time $t$ to achieve the density profile $\rho(x, \tau)$ at time $\tau$ is

$$j(x, \tau) = q(x) = -D\big(\rho(x, \tau)\big)\frac{d\rho(x, \tau)}{dx} + \sigma\big(\rho(x, \tau)\big)\left(\frac{d}{dx}\frac{\delta\mathcal{F}_\tau(\rho(x, \tau))}{\delta\rho(x, \tau)}\right) \tag{134}$$

and that the evolution of $\mathcal{F}_\tau$ satisfies the following Hamilton-Jacobi equation

$$\frac{d\mathcal{F}_\tau}{d\tau} = \int_0^1 dx \left(\frac{d}{dx}\frac{\delta\mathcal{F}_\tau}{\delta\rho(x)}\right)\left[D(\rho(x))\rho'(x) - \frac{\sigma(\rho(x))}{2}\left(\frac{d}{dx}\frac{\delta\mathcal{F}_\tau}{\delta\rho(x)}\right)\right] . \tag{135}$$

## 11.2 The equations satisfied by optimal trajectory

For large $L$ there are optimal trajectories $\{\rho(x,\tau'), j(x,\tau')\}$ for $\tau_0 < \tau' < \tau$ which maximize (131) under the constraints that $\rho(x,\tau_0)$ and $\rho(x,\tau)$ are fixed and that the conservation law (114) is satisfied. These optimal trajectories can be found [13] by solving the following coupled equations for $\tau_0 < \tau' < \tau$

$$
\begin{aligned}
\frac{d\rho}{d\tau'} &= \frac{d}{dx}\left(D(\rho)\frac{d\rho}{dx}\right) - \frac{d}{dx}\left(\sigma(\rho)\frac{dH}{dx}\right) \\
\frac{dH}{d\tau'} &= -D(\rho)\frac{d^2H}{dx^2} - \frac{\sigma'(\rho)}{2}\left(\frac{dH}{dx}\right)^2
\end{aligned}
\tag{136}
$$

where the densities at the boundaries are fixed by the reservoirs ($\rho(0,\tau') = \rho_1, \rho(1,\tau') = \rho_2$), the field $H(x,\tau')$ vanishes at the boundaries ($H(0,\tau') = H(1,\tau') = 0$)), and its initial condition $H(x,\tau_0)$ is such that the solution of (136) leads to the desired density profile $\rho(x,\tau)$ at the final time $\tau$.

These coupled equations (136) have been derived by several methods (see for example [13, 48,130] and the Appendix B which reproduces a calculation of [30]). Although the calculation of the large deviation function of the density or of the current can be reduced to solving these coupled equations for various initial and final conditions, they are usually too hard to solve. In fact, so far there are only a limited number of cases for which they have been solved (see section 14).

Comparing (136) with (134) and (B-4) we see that

$$
H(x,\tau') = \frac{\delta\mathcal{F}_\tau\big(\rho(x,\tau)\big)}{\delta\rho(x,\tau)} \quad .
\tag{137}
$$

**Remark:** the Hamilton-Jacobi (135) and the equations for the optimal profile (137) can easily be generalized to higher dimension. They become for isotropic systems in a $d$-dimension domain $\mathbb{D}$

$$
\frac{d\mathcal{F}_{\tau'}}{d\tau'} = \int_{\mathbb{D}} dx \left(\nabla\frac{\delta\mathcal{F}_{\tau'}}{\delta\rho(x)}\right)\cdot\left[D\big(\rho(x)\big)\nabla\rho(x) - \frac{\sigma\big(\rho(x)\big)}{2}\left(\nabla\frac{\delta\mathcal{F}_{\tau'}}{\delta\rho(x)}\right)\right]
$$

$$
\partial_{\tau'}\rho = \nabla\cdot(D(\rho)\nabla\rho) - \nabla\cdot(\sigma(\rho)\nabla H)
$$

$$
\partial_{\tau'}H = -D(\rho)\Delta H - \frac{1}{2}\sigma'(\rho)(\nabla H)^2 \quad .
$$

# 12 How to relate the microscopic and the macroscopic scales

In this section, we show, in the simple case of non-interacting random walkers on a ring of $L$ sites, how to relate the large scale description based on the macroscopic fluctuation theory to the microscopic calculation.

## 12.1 The macroscopic calculation

Given an initial profile $\rho(x,\tau_0)$ let us define $G_\tau\big(\{A(x), B(x)\}\big)$ by

$$
G_\tau = \frac{1}{L}\log\left\langle\exp\left[L\int_0^1 dx A(x)\rho(x,\tau) + L\int_0^1 dx B(x)\int_{\tau_0}^\tau d\tau' j(x,\tau')\right]\right\rangle
$$

where $\langle . \rangle$ means an average over the measure (130). For large $L$ one has

$$G_\tau = \max_{\{\rho(x,\tau'),j(x,\tau')\}} \left[ \int_0^1 dx \left( A(x)\rho(x,\tau) + \int_{\tau_0}^\tau d\tau' \left[ B(x)j - \frac{(j+D(\rho)\rho')^2}{2\sigma(\rho)} \right] \right) \right] \quad .$$

Using the fact that $\rho(x,\tau+\delta\tau) = \rho(x,\tau) + \delta\rho(x) = \rho(x,\tau) - \partial_x j(x,\tau)\delta\tau$ one gets after an integration by part

$$\frac{dG_\tau}{d\tau} = \max_{\tilde{j}(x)} \int_0^1 dx \left[ (B(x) + A'(x))\tilde{j} - \frac{(\tilde{j} + D(\tilde{\rho})\tilde{\rho}')^2}{2\sigma(\tilde{\rho})} \right]$$

where

$$\tilde{j}(x) = j(x,\tau) \quad \text{and} \quad \tilde{\rho}(x) = \frac{\delta G_\tau}{\delta A(x)} \tag{138}$$

and after an integration over $\tilde{j}$ this leads to

$$\frac{dG_\tau}{d\tau} = \int_0^1 dx \, \frac{\sigma(\tilde{\rho}(x))(A'(x) + B(x))^2}{2} - (A'(x) + B(x))D(\tilde{\rho}(x))\tilde{\rho}'(x) \tag{139}$$

where $\tilde{\rho}(x)$ is given by (138).

So far the transport coefficients $D(\rho)$ and $\sigma(\rho)$ are arbitrary in this macroscopic calculation.

## 12.2 The microscopic calculation

Let us consider now one specific microscopic model, the case of non-interacting random walkers on a ring. On a ring of $L$ sites, a microscopic configuration is specified by the numbers of particles $n_i$ of particles on each site $1 \le i \le L$ (because of periodic boundary conditions $n_{i+L} = n_i$). During every infinitesimal time interval $dt$, for each pair $i, i+1$ of nearest neighbors $i, i+1$ on the chain, there is a probability $n_i dt$ that $(n_i, n_{i+1}) \to (n_i - 1, n_{i+1} + 1)$ and a probability $n_{i+1} dt$ that $(n_i, n_{i+1}) \to (n_i + 1, n_{i+1} - 1)$. Let us define the following generating function

$$\Psi(t) = \left\langle \exp\left[ \sum_{i=1}^L \alpha_i n_i(t) + \beta_i Q_i(t) \right] \right\rangle \tag{140}$$

where $Q_i(t)$ is the integrated current between sites $i$ and $i+1$ during time $t$. It is easy to see that

$$\begin{aligned}
\frac{d\Psi(t)}{dt} &= \sum_i \left\langle n_i(t)\left( e^{\alpha_{i+1}-\alpha_i+\beta_i} + e^{\alpha_{i-1}-\alpha_i-\beta_{i-1}} - 2 \right) \exp\left[ \sum_{i=1}^L \alpha_i n_i(t) + \beta_i Q_i(t) \right] \right\rangle \\
&= \sum_i \left( e^{\alpha_{i+1}-\alpha_i+\beta_i} + e^{\alpha_{i-1}-\alpha_i-\beta_{i-1}} - 2 \right) \partial_{\alpha_i} \Psi(t) \quad .
\end{aligned} \tag{141}$$

For large $L$, let us choose for $\alpha_i$ and $\beta_i$ slowly varying functions of $i$ of the form

$$\alpha_i = A\left(\frac{i}{L}\right) \quad ; \quad \beta_i = \frac{1}{L}B\left(\frac{i}{L}\right) \tag{142}$$

and assume that

$$\frac{\log\Psi(t)}{L} = \tilde{G}_t(\{A(x), B(x)\}) \quad .$$

Using the fact that $\partial_{\alpha_i} \simeq \frac{1}{L}\frac{\delta}{\delta A(x)}$ for $x = i/L$, one gets

$$\frac{d\widetilde{G}_t}{dt} = \frac{1}{L^2}\int_0^1 dx\left(\left(A'(x) + B(x)\right)^2 + \left(A''(x) + B'(x)\right)\right)\widetilde{\rho}(x) \tag{143}$$

where $\widetilde{\rho}(x)$ is defined by

$$\widetilde{\rho}(x) = \frac{\delta\widetilde{G}_t}{\delta A(x)} \quad . \tag{144}$$

If we want (138,139) to coincide with the expressions (144,143), we need (after an integration by part) that

$$D(\rho) = 1 \quad \text{and} \quad \sigma(\rho) = 2\rho$$

in agreement with (111). This shows, at least in this simple example of non-interacting random walkers, that the macroscopic description (131) based on fluctuating hydrodynamics and on the macroscopic fluctuation theory (115,131) gives a large scale description of the microscopic particle system.

**Remark:** if one repeats the same calculation in the case of the SSEP on a ring, (141) becomes

$$\frac{d\Psi}{dt} = \sum_i \left(e^{\alpha_{i+1}-\alpha_i+\beta_i} - 1\right)\left(\partial_{\alpha_i} - \partial^2_{\alpha_i,\alpha_{i+1}}\right)\Psi + \left(e^{-\alpha_{i+1}+\alpha_i-\beta_i} - 1\right)\left(\partial_{\alpha_{i+1}} - \partial^2_{\alpha_i,\alpha_{i+1}}\right)\Psi$$

and one can show that one should take $D(\rho) = 1$ and $\sigma(\rho) = 2\rho(1-\rho)$ for the microscopic calculation to coincide with (139), in agreement with (107).

## 13   Large deviation function of the density in the steady state

Consider a one dimensional lattice gas (for example the SSEP) of $L$ sites maintained in its steady state by contacts with two reservoirs of particles. If one decomposes the system into $k$ large boxes of size $l = L/k$ (as in section 9) the probability that there are $N_1$ particles in the first box, $N_2$ particles in the second box,$\cdots$, $N_k$ particles in the $k$-th box takes, for large $L$, a large deviation form

$$\mathbb{P}_{\text{steady state}}\left(\frac{N_1}{l} = r_1, \ldots, \frac{N_k}{l} = r_k\right) \sim e^{-L\mathcal{F}(r_1,\ldots,r_k)} \tag{145}$$

If the number of boxes is large (writing $l = Ldx$ with $1 \ll l \ll L$ and the number of particles in the box $(Lx, L(x + dx))$ is $L\rho(x)dx$, the rate function $\mathcal{F}$ becomes a functional of the density profile $\rho(x)$

$$\mathbb{P}_{\text{steady state}}\left(\{\rho(x)\}\right) \sim e^{-L\mathcal{F}\left(\{\rho(x)\}\right)} \tag{146}$$

Our goal in this section is to discuss the properties of this large deviation functional $\mathcal{F}\left(\{\rho(x)\}\right)$.

### 13.1   Equilibrium

Let us first consider a system at equilibrium in contact with a reservoir at density $\rho^*$. If the interactions are short range, one can write

$$\mathbb{P}_{\text{eq.}}\left(\frac{N_1}{l} = r_1, \ldots, \frac{N_k}{l} = r_k\right) \sim \frac{1}{Z_L(L\rho^*)\,e^{\mu(\rho^*)L\rho^*}}\prod_{m=1}^k Z_l(lr_m)\,e^{\mu(\rho^*)l\,r_m} \tag{147}$$

where $Z_l(N)$ is the partition function of a system of size $l$ and $N$ particles and $\mu(\rho^*)$ is the chemical potential. (In (147), the contributions of the interactions between sites belonging to different boxes have been neglected because these "surface contributions" become subdominant for large $L$). Assuming also that the boxes are large enough, so that one can use the thermodynamic limit in each box, one obtains

$$\mathbb{P}_{\text{eq.}}\left(\frac{N_1}{l} = r_1, \ldots, \frac{N_k}{l} = r_k\right) \sim \exp\left[Lf(\rho^*) - L\rho^* f'(\rho^*) + l\sum_{m=1}^{k} r_m f'(\rho^*) - f(r_m)\right]$$

where $f(r)$ is the free energy density $f(r)$ defined by $\log Z_l(lr) = -lf(r)$, and where we have used that $\mu(\rho) = f'(\rho)$ (see (82)).

Therefore, as $L = kl$, the large deviation functions defined in (145,146) are

$$\mathcal{F}(r_1, \ldots, r_k) = \frac{l}{L} \sum_{m=1}^{k} \left(f(r_m) - f(\rho^*) - (r_m - \rho^*)f'(\rho^*)\right)$$

and

$$\mathcal{F}(\{\rho(x)\}) = \int_0^1 dx \left[f(\rho(x)) - f(\rho^*) - (\rho(x) - \rho^*)f'(\rho^*)\right] . \tag{148}$$

We see that at equilibrium, for systems with short range interactions, the large deviation functional has an explicit expression in terms of the free energy. Moreover it is a *local* functional since for $x \neq y$

$$\frac{\delta^2 \mathcal{F}}{\delta\rho(x)\,\delta\rho(y)} = 0.$$

The generating functional of the density

$$\mathcal{G}(\{A(x)\}) = \frac{1}{L} \log \left\langle \exp\left[L \int_0^1 A(x)\rho(x)dx\right]\right\rangle$$

is local as well, as it is related to $\mathcal{F}$ via a Legendre transform

$$\mathcal{G}(\{A(x)\}) = \max_{\{\rho(x)\}} \left[-\mathcal{F}(\{\rho(x)\}) + \int_0^1 A(x)\rho(x)dx\right] . \tag{149}$$

## 13.2   Non-locality of the large deviation functional out of equilibrium

Consider the steady state of a general lattice gas (on a one dimensional lattice of $L$ sites). The generating function of the density defined by

$$\Psi = \left\langle \exp\left[\sum_i \alpha_i n_i\right]\right\rangle \tag{150}$$

can be expanded in powers of the $\alpha_i$'s:

$$\log \Psi = \sum_i \alpha_i \langle n_i\rangle + \frac{1}{2}\sum_{i,j} \alpha_i \alpha_j \left(\langle n_i n_j\rangle - \langle n_i\rangle\langle n_j\rangle\right) + O(\alpha^3) . \tag{151}$$

When $\alpha_i$ is a slowly varying function $\alpha_i = A\left(\frac{i}{L}\right)$ as in (142) this becomes (see (123))

$$\mathcal{G}(\{A(x)\}) = \frac{\log \psi}{L} = \int_0^1 dx\, A(x)\rho^*(x) + \frac{1}{2}\int_0^1 dx\int_0^1 dy\, A(x)A(y)\langle\rho(x)\rho(y)\rangle_c + O(A^3)$$

$$= \int_0^1 dx \left[A(x)\rho^*(x) + \frac{A(x)^2}{2f''(\rho^*(x))} + \int_0^1 dy \frac{A(x)A(y)C(x,y)}{2}\right] + O(A^3) . \tag{152}$$

We see that, in general, in a non-equilibrium steady state, $\mathcal{G}$ is non-local as soon as the long range correlations $C(x, y)$ in (122) of the density do not vanish. This implies that the large deviation functional $\mathcal{F}$

$$\mathcal{F}\big(\{\rho(x)\}\big) = \max_{\{A(x)\}} \left[ -\mathcal{G}\big(\{A(x)\}\big) + \int_0^1 A(x)\rho(x)dx \right] \tag{153}$$

is non-local as well.

For the SSEP [57], knowing the explicit expression (125,127) of the two-point correlations, (152) becomes

$$\mathcal{G}(\{A(x)\}) = \int_0^1 dx \left[ A(x)\rho^*(x) + A(x)^2 \frac{\rho^*(x)\big(1 - \rho^*(x)\big)}{2} \right]$$
$$- (\rho_1 - \rho_2)^2 \int_0^1 dx \int_x^1 dy A(x)A(y)x(1-y) + O(A^3) \tag{154}$$

with $\rho^*(x)$ given by (124).

**Remark:** as exact expressions of all the correlations are known fo the SSEP (see (99,100) and [46, 56]), one can expand $\mathcal{G}$ to arbitrary orders in powers of $A$. In fact for the SSEP the full expression of $\mathcal{G}(\{A(x)\})$ is known [57]

$$G(\{A(x)\}) = \int_0^1 dx \left[ \log\big(1 - F(x) + F(x)e^{A(x)}\big) - \log\frac{F'(x)}{\rho_2 - \rho_1} \right] \tag{155}$$

where $F(x)$ is the monotonic function solution of

$$F'' + \frac{F'^2\big(1 - e^{A(x)}\big)}{1 - F + F e^{A(x)}} = 0$$

which satisfies $F(0) = \rho_1$ and $F(1) = \rho_2$.

## 14 The macroscopic fluctuation theory and the large deviation functional of the density

According to the macroscopic fluctuation theory [13, 14, 17, 19], in the steady state, the large deviation functional of the density can be obtained as the minimal cost (132)

$$\mathcal{F}\big(\{\rho(x)\}\big) = \min_{\{\rho(x,\tau'), j(x,\tau')\}} \left[ \int_0^1 dx \int_{-\infty}^{\tau} d\tau' \frac{(j + D(\rho)\rho')^2}{2\sigma(\rho)} \right] \tag{156}$$

to observe a density profile $\rho(x)$ at time $\tau$, starting at time $-\infty$ from the steady state profile $\rho(x, -\infty) = \rho^*(x)$ (see (118)). To do so one needs to solve the equations (136) with these conditions on the density at time $-\infty$ and at time $\tau$. (Note that the above expression (156) does not depend on $\tau$, physically because in the steady state, the probability of observing a density profile $\rho(x)$ does not depend on the time $\tau$ at which this profile is observed, and mathematically because (156) remains unchanged by varying the time $\tau$ as the density and current profiles which minimize (156) are functions of the difference $\tau' - \tau$). As shown in the

appendix (B-6), the large deviation functional can also be written in terms of the functions $\rho$ and $H$ solutions of the equations (136) as

$$\mathcal{F}\big(\{\rho(x)\}\big) = \int_{-\infty}^{\tau} d\tau' \int_0^1 dx \frac{\sigma(\rho)}{2} \left(\frac{dH}{dx}\right)^2 \quad . \tag{157}$$

So one way to calculate the functional $\mathcal{F}$ is first, to solve the equations (136), and then, to use (157).

For general transport coefficients $D(\rho)$ and $\sigma(\rho)$, the solution of (136) and the explicit expression of the large deviation functional $\mathcal{F}(\{\rho(x)\})$ or of the generating function $\mathcal{G}$ are not known. As we will see below, there are however a few cases where this expression is known.

**Remark:** in some cases [3,6,18,34,35], varying parameters as the densities of the reservoirs or in presence of an external field, there might be several trajectories $\{\rho(x, \tau'), j(x, \tau')\}$ which minimize (156) giving rise to phase transitions where the functional $\mathcal{F}$ is non-analytic. (This is very reminiscent of the possibility of multiple optimal trajectories in the principle of least action.)

## 14.1   The optimal profiles at equilibrium.

We have seen (148) that at equilibrium $\mathcal{F}$ is known. Then using the fact that $\partial_x \rho^*(x) = 0$, one can show that the solution of the equations (136) is

$$\partial_{\tau'} \rho(x, \tau') = -\partial_x \Big( D\big(\rho(x, \tau')\big) \partial_x \rho(x, \tau') \Big) \tag{158}$$

with

$$H(x, \tau') = f'\big(\rho(x, \tau')\big) - f'\big(\rho^*(x)\big) \tag{159}$$

and the boundary condition $\rho(x, \tau) = \rho(x)$.

One can rewrite (158) as

$$\rho(x, \tau') = \varphi(x, 2\tau - \tau') \quad \text{where} \quad \partial_s \varphi = \partial_x \Big( D(\varphi) \partial_x \varphi \Big) \quad . \tag{160}$$

Under this change of variable $\rho \to \varphi$, we see that $\varphi(x, s)$ for $s \in (\tau, \infty)$ is the relaxation profile starting at time $s = \tau$ with the initial profile $\varphi(x, \tau) = \rho(x)$.

So the excitation trajectory $\rho(x, \tau')$ for $\tau' \in (-\infty, \tau)$ is precisely the time reversal of the relaxation trajectory $\varphi(x, s)$ followed by the system for $s \in (\tau, \infty)$ to relax from the density profile $\rho(x)$ at time $\tau$ to the steady state profile $\rho^*(x)$.

**Remark:** close to equilibrium the difference between this excitation trajectory and this relaxation trajectory is small and it is possible to implement a perturbation calculation [30] to determine $\mathcal{F}$.

## 14.2   The optimal profile for the Zero Range process

For the Zero Range Process [13], the transport coefficients are known (110) in terms of the free energy: $D(\rho) = f''(\rho) e^{f'(\rho)}$ and $\sigma(\rho) = 2e^{f'(\rho)}$. It is then possible to check that the steady state profile $\rho^*(x)$ satisfies (118)

$$\exp[f'(\rho^*(x))] = (1-x)\exp[f'(\rho_1)] + x \exp[f'(\rho_2)] \tag{161}$$

that the solution of (136) is given by

$$H(\rho, \tau') = f'\big(\rho(x, \tau')\big) - f'\big(\rho^*(x)\big)$$

where $\rho(x, \tau')$ evolves according to

$$\partial_{\tau'}\rho = -\partial_x\big(D(\rho)\partial_x\rho\big) + \partial_x\big(\sigma(\rho)f''(\rho^*)\,\partial_x\rho^*\big) \tag{162}$$

and that

$$\mathcal{F}\{\rho(x)\}\} = \int_0^1 \Big[f\big(\rho(x)\big) - f\big(\rho^*(x)\big) - \big(\rho(x) - \rho^*(x)\big)f'\big(\rho^*(x)\big)\Big]dx \quad . \tag{163}$$

Comparing (162) with (158) we see that, because out of equilibrium $\partial_x\rho^*(x) \neq 0$, the excitation path is no longer the time reversal of the relaxation path. Still, for the Zero Range Process, the large deviation functional (163) is local [13].

## 14.3 The large deviation functional for the SSEP

We have seen in section 13.2 that for general transport coefficients $D(\rho)$ and $\sigma(\rho)$, the large deviation functional $\mathcal{F}$ is non-local. So far there are only a few exactly soluble cases for which this non-local expression of $\mathcal{F}$ is known. Some expressions can be derived either starting from the knowledge of the steady state measure of the microscopic model microscopic model as for the SSEP, the WASEP, the KMP [38, 43, 51, 52, 63, 129, 130] or by solving the equations (136) [13, 19–21]. In the case of the SSEP [13, 51, 52] the large deviation functional can be written as

$$\mathcal{F}(\rho) = \max_{F(x)}\left[\int_0^1 dx\left(\rho(x)\log\frac{\rho(x)}{F(x)} + \big(1 - \rho(x)\big)\log\frac{1 - \rho(x)}{1 - F(x)} + \log\frac{F'(x)}{\rho_2 - \rho_1}\right)\right] \tag{164}$$

where the maximum is over all monotonous functions $F(x)$ which satisfy $F(0) = \rho_1$ and $F(1) = \rho_2$. It is then easy to verify that, given a profile $\rho(x)$, the optimal function $F(x)$ in (164) is solution of

$$\rho = F + \frac{F(1 - F)F''}{F'^2} \quad . \tag{165}$$

One way to check that (164) is indeed the right expression is to verify that

$$\rho(x, \tau') = F(x, \tau') + \frac{F(x, \tau')\big(1 - F(x, \tau')\big)F''(x, \tau')}{F(x, \tau')'^2}$$

$$H(x, \tau') = \log\left[\frac{\rho(x, \tau')\big(1 - F(x, \tau')\big)}{F(x, \tau')\big(1 - \rho(x, \tau')\big)}\right]$$

are solution of (136) when $F(x, \tau')$ is the solution of

$$\partial_{\tau'}F(x, \tau') = -\partial_{xx}F(x, \tau')$$

for $-\infty < \tau' \leq \tau$ with $F(0, \tau') = \rho_1$ and $F(1, \tau') = \rho_2$. Due to these boundary conditions, one also has $F''(0, \tau') = F''(1, \tau') = 0$ and this allows to show that (164) is a stationary solution of the Hamilton-Jacobi equation (135).

**Remark:** note that the function $F(x)$ solution of (165) is in general a non-local functional of the density. For example for $\rho_1 - \rho_2$ small,

$$F(x) = \rho^*(x) - \frac{(\rho_1 - \rho_2)^2}{\rho_1(1 - \rho_1)}\left[x\int_x^\infty (1 - y)\big(\rho(y) - \rho^*(y)\big)dy\right.$$

$$\left. + (1 - x)\int_0^x y\big(\rho(y) - \rho^*(y)\big)\right] + O\big((\rho_1 - \rho_2)^3\big)$$

**Remark:** as discussed at the end of section 13 the alternative way of obtaining (164) is to start from the exact knowledge of the steady state measure $P_{\text{steady state}}(\mathcal{C})$ which is known for the SSEP [51,52]. This approach allows to go further than the derivation of (164) as one can determine finite size corrections [58] to (164) (i.e. prefactors of (146)).

For general transport coefficients $D(\rho)$ and $\sigma(\rho)$, the explicit expression of the large deviation functional $\mathcal{F}(\{\rho(x)\})$ or of the generating function $\mathcal{G}$ is not known. There are however a few cases, in addition to the SSEP [49,51,52,130] either because one has an explicit knowledge of the steady state measure of the microscopic model as in [36–38,63,75] or because the equations (136) can be solved as in [20,117].

## 15 The macroscopic fluctuation theory and the large deviation function of the current

Consider a one dimensional diffusive system of size $L$ in contact with two reservoirs of particles at densities $\rho_1$ and $\rho_2$ as in Figure 5. Let $Q_t$ be the flux of particles from reservoir **1** to reservoir **2** during time $t$. As in section 6.2, we assume that the particles cannot accumulate indefinitely in the system, so that for large $t$, the ratio $\frac{Q_t}{t}$ does not depend on where the flux is measured.

Due to the rescaling of the current (117) and of the time $t = L^2 \tau$ between the microscopic and the macroscopic description, the probability that $\frac{Q_t}{t} = \frac{q}{L}$ is given according to the macroscopic fluctuation theory by

$$\mathbb{P}\left(\frac{Q_t}{t} = \frac{q}{L}\right) \sim e^{-\frac{t}{L} I(q)} = e^{-\tau L I(q)} \tag{166}$$

where

$$I(q) = \lim_{\tau \to \infty} \left(\frac{1}{\tau} \min_{\{\rho(x,\tau'), j(x,\tau')\}} \left[\int_0^1 dx \int_0^\tau d\tau' \frac{(j + \rho' D(\rho))^2}{2\sigma(\rho)}\right]\right) \tag{167}$$

$\tau = t/L^2$, and $j(x, \tau')$ should satisfy the following constraint

$$\int_0^\tau j(x, \tau') d\tau' = q\,\tau \tag{168}$$

in addition to the conservation law (114).

**Remark:** it is easy to see that if $\rho(x, \tau')$ and $j(x, \tau')$ are the optimal profiles giving $I(q)$, in (167), then $\widetilde{\rho}(x, \tau') = \rho(x, \tau - \tau')$ and $\widetilde{j}(x, \tau') = -j(x, \tau - \tau')$ are the optimal profiles giving $I(-q)$ and this implies that

$$I(q) - I(-q) = \lim_{\tau \to \infty}\left(\frac{2}{\tau}\int_0^1 dx \int_0^\tau j\frac{\rho' D(\rho)}{\sigma(\rho)}\right) = q\int_{\rho_1}^{\rho_2}\frac{2D(\rho)}{\sigma(\rho)} d\rho \quad .$$

This can be rewritten (see (82, 89)) as

$$I(q) - I(-q) = q\left(f'(\rho_2) - f'(\rho_1)\right) = q\left(\mu(\rho_2) - \mu(\rho_1)\right)$$

which is nothing but the Fluctuation Theorem (63).

**Remark :** The macroscopic fluctuation theory is not the only way to determine the large deviation function of the current. Fore a number of exactly soluble cases [4, 81, 101] the largest eigenvalue of the tilted matrix (3.1) can also be calculated exactly for arbitrary system sizes.

# 16 Large deviations of the current assuming time-independent optimal profiles

In general, the optimal density and current profiles $\rho(x, \tau')$ and $j(x, \tau')$ which minimize (167) are time dependent. In 2004, however, in a joint work with Thierry Bodineau [26], we obtained an expression of the large deviation function $I(q)$ assuming that the optimal density and current profiles are time independent. (This hypothesis is called the additivity principle [26] because it allows to relate the large deviation of the current of a system between reservoirs at densities $\rho_1$ and $\rho_2$ to those of two subsystems, one between reservoirs at densities $\rho_1$ and $\rho_3$ and another one between reservoirs at densities $\rho_3$ and $\rho_2$.) This hypothesis was shown to be correct by Bertini et al. [16, 19, 20] if the transport coefficients satisfy $D(\rho)\sigma''(\rho) \leq D'(\rho)\sigma'(\rho)$. This is the case for the the SSEP (97), the Zero Range Process (110), but not for the KMP model (108).

If the two profiles are time independent, it follows from the conservation law (114) that the current $j$ is also space independent, so that $j = q$. Then to obtain (167), one only needs to find the optimal time independent profile $\rho(x)$ which minimizes the following expression

$$I(q) = \min_{\{\rho(x)\}} \int_0^1 \frac{(q + \rho' D(\rho))^2}{2\sigma(\rho)} dx \quad . \tag{169}$$

This is a Lagrangian problem, with the Lagrangian density

$$\mathcal{L}(\rho, \rho') = \frac{(q + \rho' D(\rho))^2}{2\sigma(\rho)} \tag{170}$$

and boundary conditions $\rho(0) = \rho_1, \rho(1) = \rho_2$. Using the associated Euler-Lagrange equations $\partial_x \left( \frac{\delta \mathcal{L}}{\delta \rho'(x)} \right) = \frac{\delta \mathcal{L}}{\delta \rho(x)}$ one can show, as usual, that $-\mathcal{L} + \rho'(x) \frac{\delta \mathcal{L}}{\delta \rho'(x)}$ does not depend on $x$. This yields the equation satisfied by the optimal density profile $\rho(x)$

$$\boxed{\left( D(\rho) \partial_x \rho \right)^2 = q^2 \left( 1 + 2K\sigma(\rho) \right)} \tag{171}$$

where $q^2 K$ plays the role of the energy in Lagrangian-Hamiltonian mechanics. For $\rho_1 > \rho_2$, a solution of (171) is $D(\rho)\partial_x \rho = -q\sqrt{1 + 2K\sigma(\rho)}$ and recognizing a total derivative in $\rho$, one can obtain [26], by varying $K$, the following parametric expression for $I(q)$ as a function of $q$ by varying $K$

$$q = \int_{\rho_2}^{\rho_1} \frac{D(\rho) d\rho}{\sqrt{1 + 2K\sigma(\rho)}} \quad ; \quad I(q) = q \int_{\rho_2}^{\rho_1} \frac{D(\rho) d\rho}{\sigma(\rho)} \left( \frac{1 + K\sigma(\rho)}{\sqrt{1 + 2K\sigma(\rho)}} - 1 \right) \quad . \tag{172}$$

In the $K \to 0$ limit one gets that $I(q) = 0$ for $q = j^*$ (see (118)). By expanding in powers of $K$ and then by eliminating $K$, one can find the expansion of $I(q)$ around its maximum at $q = j^*$.

$$I(q) = \frac{(q - j^*)^2}{2S_2} + \frac{(S_2^2 - S_1 S_3)(q - j^*)^3}{2S_2^3} + \mathcal{O}\left( (q - j^*)^4 \right) \tag{173}$$

where

$$S_n = \int_{\rho_2}^{\rho_1} \sigma^{n-1}(\rho) D(\rho) d\rho \quad ; \quad j^* = S_1 \quad .$$

Alternatively, one can consider the generating function $\langle e^{\lambda Q_t} \rangle$. From (166) one has

$$\langle e^{\lambda Q_t} \rangle \sim \exp\left[ \frac{\mu(\lambda) t}{L} \right]$$

with $\mu(\lambda)$ given by $\mu(\lambda) = \max_q[\lambda q - I(q)]$. Using (173) one can expand $\mu(\lambda)$ in powers of $\lambda$

$$\mu(\lambda) = S_1\lambda + \frac{S_2}{2S_1}\lambda^2 + \frac{S_1 S_3 - S_2^2}{2S_1^3}\lambda^3 + \mathcal{O}(\lambda^4)$$

allowing to determine this way all the cumulants of the current (see (36) and [26, 28])

$$\lim_{t\to\infty}\frac{\langle Q_t\rangle}{t} = \lim_{\tau\to\infty} = \frac{S_1}{L} \quad ; \quad \lim_{t\to\infty}\frac{\langle Q_t^2\rangle_c}{t} = \frac{S_2}{S_1 L} \quad ; \quad \ldots \tag{174}$$

Note that in the limit $\rho_2 \to \rho_1$ one recovers (88).

**Remark:** The expressions (172) were obtained under the assumption that the optimal time independent profile $\rho(x)$ solution of (171) is a monotonic decreasing function (for $\rho_1 > \rho_2$). If $\sigma(\rho)$ has no maximum between $\rho_2$ and $\rho_1$, the current $q$ in (172) reaches a maximum finite value as $K \to -\frac{1}{2\sigma(\rho_1)}$ (assuming that $\sigma(\rho_1) > \sigma(\rho_2)$ and $\sigma'(\rho_1) > 0$). To go beyond this maximal value of $q$, one needs to allow the optimal profile to be non-monotonic. For example if $\sigma(\rho)$ is an increasing function of $\rho$ (as in the KMP model (108)), the expression (172) takes the following parametric form by varying $\rho_0$ between $\rho_1$ and $\infty$

$$q = \int_{\rho_1}^{\rho_0} \frac{D(\rho)d\rho}{\sqrt{1+2K\sigma(\rho)}} + \int_{\rho_2}^{\rho_0} \frac{D(\rho)d\rho}{\sqrt{1+2K\sigma(\rho)}} \quad \text{with} \quad K = -\frac{1}{2\sigma(\rho_0)} \tag{175}$$

$$I(q) = q\left[\int_{\rho_1}^{\rho_0} \frac{D(\rho)d\rho}{\sigma(\rho)}\frac{1+K\sigma(\rho)}{\sqrt{1+2K\sigma(\rho)}} + \int_{\rho_2}^{\rho_0} \frac{D(\rho)d\rho}{\sigma(\rho)}\frac{1+K\sigma(\rho)}{\sqrt{1+2K\sigma(\rho)}} - \int_{\rho_2}^{\rho_1} \frac{D(\rho)d\rho}{\sigma(\rho)}\right] . \tag{176}$$

When $\sigma(\rho)$ has a (quadratic maximum) at some value $\rho^*$ with $\rho_2 < \rho^* < \rho_1$ one can see from (172) that $q$ can vary from $0$ to $\infty$ as $K$ varies from $-\frac{1}{2\sigma(\rho^*)}$ to $\infty$. On the other hand if $\sigma(\rho)$ takes values larger than $\sigma(\rho^*)$ outside the interval $(\rho_1, \rho_2)$, one can obtain alternative expressions like (175,176) (associated to non-monotonic profiles) which could give values of $I(q)$ smaller than (172) for some range of $q$.

In general, if $\sigma(\rho)$ has a complicated form with more than one maximum in or outside the interval $(\rho_1, \rho_2)$, several optimal profiles might lead in (172,175) to the same value of $q$ but different values of $I(q)$ (see next section).

**Remark:** For higher dimensional diffusive systems in contact with two reservoirs one can also find the optimal time-independent density and current profiles [2] which minimize the $d$-dimensional version of (169).

## 17 Phase Transitions in the large deviation function of the current

Since the above parametric expressions of $I(q)$ are known to be correct only under some conditions on the transport coefficients $D(\rho)$ and $\sigma(\rho)$, it is a priori possible that they could be valid only in certain ranges of values of $q$. This would imply that as $q$ varies, the large deviation function $I(q)$ may undergo phase transitions.

A first possibility would simply be that there are several time independent profiles which make the expression (169) stationary and that as $q$ varies, there is a jump of the optimal profile from one of these stationary profiles to another one [7, 8].

A second possibility would be that, for some transport coefficients $D(\rho)$ and $\sigma(\rho)$ and some values of $q$, the above expressions (172,175,176) of $I(q)$ are not convex (see section 5). If so, the true $I(q)$ could simply be the convex envelope of the above expressions [15, 16].

A third possibility is that as $q$ varies, the optimal density and current profiles become time dependent [27, 28, 64, 84–87, 122]. In this case, if the transition from a time-independent to a time-dependent profile is second order, one can try to predict the location of the phase transition. The simplest situation is the case of a diffusive system on a ring (of macroscopic length 1 with $L\bar{\rho}$ particles). Near such a second order dynamical phase transition, the time variations of the optimal profiles should be small, of the form .

$$\rho = \bar{\rho} + \varepsilon \sin(2\pi(x - vt)) \tag{177}$$

which implies, by the conservation equation,

$$j = q + \varepsilon v \sin(2\pi(x - vt)) \quad . \tag{178}$$

At order $\epsilon^2$ the expression (167) becomes

$$I(q) = \frac{q^2}{2\sigma(\bar{\rho})} + \epsilon^2 \left[ \frac{\left(\sigma(\bar{\rho})v - q\sigma'(\bar{\rho})\right)^2}{4\sigma(\bar{\rho})^3} + \frac{8\pi^2 \sigma(\bar{\rho}) D(\bar{\rho})^2 - q^2 \sigma''(\bar{\rho})}{8\sigma(\bar{\rho})^2} \right] \quad .$$

We see that, by choosing $v = q\frac{\sigma'(\bar{\rho})}{\sigma(\bar{\rho})}$, if

$$8\pi^2 \sigma(\bar{\rho}) D(\bar{\rho}) - q^2 \sigma''(\bar{\rho}) < 0 \tag{179}$$

time-dependent profiles lead to a lower $I(q)$ than the static profiles $\rho(x) = \bar{\rho}$. This means that if (179) is satisfied, the flat profile $\bar{\rho}$ is unstable. Note that for concave $\sigma(\rho)$ as in the SSEP, (179) can never be satisfied.

Because (179) is a consequence of a perturbative analysis around a time independent profile, one cannot exclude a first order phase transition to occur before this condition is satisfied.

**Remark:** in the simple case of the ring geometry, when the optimal profile $\rho(x) = \bar{\rho}$ is flat, only the second cumulant of $Q_t$ is of order $1/L$ and $\mu(\lambda) = \sigma(\bar{\rho}))\lambda^2/2$. By computing the higher cumulants at order $\frac{1}{L^2}$ the condition (179) reappears as a singularity of the generating function of the cumulants [4, 123].

**Remark:** these dynamical phase transitions where a flat profile becomes unstable are very reminiscent of the phase transition of other non-equilibrium systems such as the ABC model or the Katz Lebowitz Spohn model [29,39,67,68,73,91,92]. Note that for these non equilibrium systems, phase transitions may occur in one dimension in contrast with systems at equilibrium with short range interactions.

## 18 Non-steady state situations: the case of the infinite line

A diffusive system on the infinite line is one of the simplest non-steady state situations one may consider, for example as in figure 10, with an initial condition where the density is $\rho_a$ on the left and $\rho_b$ on the right of the origin. [47, 48, 78, 108, 114, 116, 121].

$$\rho(x, \tau = 0) = \begin{cases} \rho_a & \text{for} \quad x < 0 \\ \rho_b & \text{for} \quad x > 0 \end{cases} \tag{180}$$

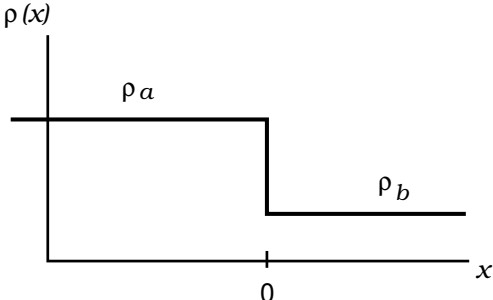

Figure 10: On the infinite line, for a diffusive system with a given initial density profile $\rho(x)$, we are interested in the large deviation function of the integrated current through the origin.

For such an initial condition, one can show that the influence of the discontinuity of the density at the origin spreads over a region of size which grows like $\sqrt{t}$ (of course this is expected for diffusive systems). For example when $D(\rho) = 1$, the average density which evolves according to $\frac{\partial \rho}{\partial t} = \frac{\partial^2 \rho}{\partial x^2}$ is given by

$$\langle \rho(x, t) \rangle = \rho_b + \frac{\rho_a - \rho_b}{\sqrt{\pi}} \int_{x/(2\sqrt{t})}^{\infty} e^{-y^2} dy$$

implying that the average integrated current through the origin is

$$\langle Q_t \rangle = \int_0^{\infty} \left( \rho(x, t) - \rho_b \right) dx = (\rho_a - \rho_b) \frac{\sqrt{t}}{\sqrt{\pi}} \quad .$$

As the average integrated current scales like $\sqrt{t}$, the current itself $\langle J_t \rangle$ scales like $\frac{1}{\sqrt{t}}$. In fact all the cumulants of the integrated current $Q_t$ scale like $\sqrt{t}$ and its large deviation function is of the form

$$\mathbb{P}\left( \frac{Q_t}{\sqrt{t}} = q \right) \asymp e^{-\sqrt{t} I(q)} \quad . \tag{181}$$

Notice the $\sqrt{t}$ scaling instead of the $t/L$ we had (174) in the case of the steady state of systems in contact with two reservoirs (this $\sqrt{t}$ is rather natural as, for a diffusive system, the distance scales like the square root of the time).

For a general diffusive systems with arbitrary transport coefficients $D(\rho)$ and $\sigma(\rho)$ one does not know how to calculate the large deviation function $I(q)$. Within the macroscopic fluctuation theory, this boils down to solving the equations (136) with, in addition to the initial condition (180), the following final condition

$$q\sqrt{t} = \int_0^{\infty} \left( \rho(x, t) - \rho(x, 0) \right) dx \quad . \tag{182}$$

In fact an initial condition such as (180) on the infinite line shown in figure 10 may actually correspond to several different situations [48], for example the **quenched** and the **annealed** initial conditions. Each of these situations leads to a different large deviation function $I(q)$ in (181).

- In the quenched case, the density in the initial condition is precisely the density given by (180). This density is fixed.

- In the annealed case, one considers that on each side of the origin, the initial condition is an equilibrium distribution at density $\rho_a$ on the left and $\rho_b$ on the right. The main difference with the quenched case is that, in the annealed case, the initial condition can fluctuate (see (187) below), so that a contribution to $I(q)$ comes from the deviations of the initial profile $\rho(x,0)$ from (180).

In general, as we will see it in the following example, one has

$$I_{\text{annealed}}(q) \leq I_{\text{quenched}}(q)$$

or equivalently for $\mu(\lambda) = \max_q [\lambda q - I(q)]$

$$\mu_{\text{quenched}}(\lambda) \leq \mu_{\text{annealed}}(\lambda) \quad \text{where} \quad \mu(\lambda) = \lim_{t \to \infty} \frac{\log \langle e^{\lambda Q_t} \rangle}{\sqrt{t}} \quad . \tag{183}$$

To illustrate this difference, let us consider the simple example of non-interacting diffusive particles in the case where $\rho_a \neq 0$ and $\rho_b = 0$. If a particle is at position $x < 0$ at time $0$, the probability $p(x,t)$ that it contributes to the integrated current $Q_t$ across the origin is

$$p(x,t) = \mathbb{P}(\text{the particle starting at } x \text{ is on the positive side at time } t)$$

and it is given by

$$p(x,t) = g\left(\frac{x}{\sqrt{t}}\right) \quad \text{where} \quad g\left(\frac{x}{\sqrt{t}}\right) = \int_{-\infty}^{x} e^{-\frac{u^2}{4t}} \frac{du}{2\sqrt{\pi t}} \tag{184}$$

(with this normalization one has $D(\rho) = 1$).

- For the quenched case, one can start on the real axis with one particle at each position $\frac{-k}{\rho_a}$ for all $k \geq 1$. Thus

$$\langle e^{Q_t} \rangle_{\text{quenched}} = \prod_{k \geq 1} \left[ 1 + (e^\lambda - 1) g\left(\frac{-k}{\rho_a \sqrt{t}}\right) \right] \tag{185}$$

and this gives in the large $t$ limit

$$\mu_{\text{quenched}}(\lambda) = \rho_a \int_{-\infty}^{0} \log\left( 1 + (e^\lambda - 1) g(u) \right) du \quad .$$

- For the annealed case, one can consider that in the initial condition, the particles on the negative real axis are the points of a Poisson process of density $\rho_a$ and *one averages over the initial condition*

Then

$$\langle e^{\lambda Q_t} \rangle_{\text{annealed}} = \prod_{dx} \left( 1 - \rho_a dx + \rho_a dx \, p(x,t) \right)$$

$$= \exp\left[ \int_{-\infty}^{0} dx \, \rho_a (e^\lambda - 1) g\left(\frac{x}{\sqrt{t}}\right) \right] \tag{186}$$

so that

$$\mu_{\text{annealed}}(\lambda) = \rho_a (e^\lambda - 1) \int_{-\infty}^{0} g(u) du \quad .$$

Clearly, since $\log(1 + x) \leq x$ for $x \geq -1$, the inequality (183) is satisfied.

- In fact for an arbitrary fixed initial density $\rho_0(y)$ on the negative real axis, one would get

$$\left\langle e^{\lambda Q_t} | \rho_0 \right\rangle = \exp\left( \int_{-\infty}^{0} \rho_0(y) dy \log\left[ 1 + (e^{\lambda} - 1) g\left( \frac{y}{\sqrt{t}} \right) \right] \right) \quad .$$

One can then check that

$$\left\langle e^{\lambda Q_t} \right\rangle_{\text{annealed}} = \max_{\{\rho_0(y)\}} \left[ e^{-\mathcal{F}(\rho_0)} \left\langle e^{\lambda Q_t} | \rho_0 \right\rangle \right] \tag{187}$$

where $\mathcal{F}(\rho_0) = \int_{-\infty}^{0} [f(\rho(y)) - f(\rho_a) - (\rho(y) - \rho_a)f'(\rho_a)]dy$ is the free energy of the initial profile $\rho_0$ (see (148) with $f(\rho) = \rho \log \rho - \rho$ for non interacting particles (see (111))). So in the case of figure 10 (when $\rho_b = 0$), the optimal initial profile in (187) is

$$\rho_0(x) = \rho_a \exp\left[ (e^{\lambda} - 1) g\left( \frac{y}{\sqrt{t}} \right) \right] \quad . \tag{188}$$

In addition to the case of non-interacting particles, the large deviation function $I(q)$ is only known in very few cases. For example in the case of Figure 10 for the SSEP when $\rho_b = 0$ one has [47, 107, 109]

$$\left\langle e^{\lambda Q_t} \right\rangle_{\text{annealed}} = \exp\left( \sqrt{t} \int_{-\infty}^{\infty} \frac{dk}{\pi} \log\left( 1 + \rho_a(e^{\lambda} - 1)e^{-k^2} \right) \right) \tag{189}$$

(which reduces (see(184,186)) when $\rho_a$ is small).

## 19 Conclusion

These lecture notes were centered on the calculation of the large deviation functions of the density and of the current. The main tool discussed here was the macroscopic fluctuation theory. This theory has been used in a broader context, in presence of a weak field as in WASEP [21, 42, 55], for diffusive systems with dissipation i.e. where the number of particles is not conserved [11, 31, 88, 111] is not conserved, for active particles [1, 97] or for ballistic dynamics [62]. For example in presence of an external field $E$ like in the WASEP (the weakly asymmetric exclusion process) the probability (130) of observing the time evolution of a density and current profile becomes

$$\mathbb{P}\left( \{ \rho(x, \tau'), j(x, \tau') \} \right) \propto \exp\left( -L \int_{0}^{1} dx \int_{\tau_0}^{\tau} d\tau' \frac{(j + D(\rho)\rho' + E\sigma(\rho))^2}{2\sigma(\rho)} \right) \tag{190}$$

We have seen that the calculation of large deviation functions using the macroscopic fluctuation theory can be reduced to finding the solution of nonlinear PDE's (136) with appropriate boundary conditions. These equations are difficult to solve and so far their solutions are known only in a limited number of cases (see section 2.5 and [9, 10, 13, 23, 24, 79, 80, 98, 99, 107]).

Besides the macroscopic fluctuation theory, other approaches have been used to determine large deviation functions, such as the product matrix method [25, 44, 46, 52], the Bethe ansatz [4, 110] or numerical methods [74, 76, 83]. Some of these methods often work for non-diffusive driven systems such as the the ASEP or the TASEP [31, 40, 41, 44, 45, 53, 54] allowing in particular to determine some large scale properties of the Kardar Parisi Zhang equation [22, 90, 118, 133, 134].

# A    Appendix A : transport coefficients of the zero range process

This appendix explains one way of deriving the free energy per unit volume and the transport coefficients of the zero range process used in Section 8.2

For an isolated zero range process of $N$ particles on $L$ sites, the invariant measure is given by

$$\text{Pro}(n_1, \cdots n_L) = \frac{1}{Z_L(N)} \left[ \prod_{i=1}^{L} v(n_i) \right] \delta_{n_1 + \cdots n_L, N}$$

where $v(0) = 1$ for $n \geq 1$

$$v(n) = \frac{v(n-1)}{u(n)}$$

(one can easily check that this satisfies detailed balance). Therefore the partition function satisfies

$$Z_L(N) = \sum_{m \geq 0} v(m) Z_{L-1}(N-m) \quad .$$

For large $L$, if $N = L\rho$, this implies that the free energy $f(\rho)$ per unit volume is solution of

$$1 = e^{f(\rho) - \rho f'(\rho)} \sum_{m \geq 0} e^{m f'(\rho)}$$

where we have used the fact that the free energy is extensive, namely

$$\lim_{L \to \infty} \frac{\log Z_L(L\rho)}{L} = -f(\rho) \quad .$$

One can then see that

$$\langle u(n) \rangle = \sum_{m \geq 1} \frac{Z_{L-1}(N-m) v(m) u(m)}{Z_L(N)} = \sum_{m \geq 0} \frac{Z_{L-1}(N-1-m) v(m)}{Z_L(N)} = \frac{Z_L(N-1)}{Z_L(N)} \to e^{f'(\rho)} \quad .$$

Therefore using (102,103,109) one has

$$D(\rho) = f''(\rho) \, e^{f'(\rho)}$$

and from (89)

$$\sigma(\rho) = 2 \, e^{f'(\rho)}$$

as claimed in (110).

# B    Appendix B : Equations to solve to find the optimal trajectory in section 11.2

This appendix reproduces a derivation [30] of the equations (136) satisfied by the optimal density and current profiles which minimize

$$\mathcal{A} = \int_{\tau_0}^{\tau} d\tau' \int_0^1 dx \, \frac{(j + D(\rho)\rho')^2}{2\sigma(\rho)} \quad . \tag{B-1}$$

One way to introduce the field $H(x, \tau)$ is to consider first an arbitrary trajectory of the density and current profiles $\{\rho(x, \tau'), j(x, \tau')\}$ for $\tau_0 \leq \tau' \leq \tau$ connecting the given profiles at times $\tau_0$ and and $\tau$. One can then define $H(x, \tau')$ as the solution of the diffusion equation

$$\frac{d}{dx} \left( j + D(\rho) \frac{d\rho}{dx} \right) = \frac{d}{dx} \left( \sigma(\rho) \frac{dH}{dx} \right) \tag{B-2}$$

with the boundary conditions.

$$H(0, \tau') = H(1, \tau') = 0 \quad . \tag{B-3}$$

Integrating (B-2) we gets an integration constant $C(t')$

$$j + D(\rho)\frac{d\rho}{dx} = \sigma(\rho)\frac{dH}{dx} + C(\tau') \tag{B-4}$$

and (B-1) becomes after an integration by part and using (B-3)

$$\mathcal{A} = \int_{\tau_0}^{\tau} d\tau' \int_0^1 dx \left[ \frac{\sigma(\rho)}{2} \left( \frac{dH}{dx} \right)^2 + \frac{C(\tau')^2}{2\sigma(\rho)} \right]$$

Clearly, one should choose $C(\tau') = 0$ to minimize $\mathcal{A}$. This establishes the first equation of (136) and

$$\mathcal{A} = \int_{\tau_0}^{\tau} d\tau' \int_0^1 dx \frac{\sigma(\rho)}{2} \left( \frac{dH}{dx} \right)^2 \tag{B-5}$$

So far, the trajectory of the density profile $\rho(x, \tau')$ is arbitrary. We now want this trajectory to be optimal. Let us look at the variation $\mathcal{A}$ due to a small variation of $\rho(x, \tau')$ due to a small variation of $H(x, \tau')$

$$(\rho, H) \quad \rightarrow \quad (\rho + \varepsilon r, H + \varepsilon h)$$

At linear order the first equation (136) becomes

$$\frac{dr}{d\tau'} = \frac{d^2(D(\rho)r)}{dx^2} - \frac{d}{dx} \left( \sigma'(\rho)r\frac{dH}{dx} \right) - \frac{d}{dx} \left( \sigma(\rho)\frac{dh}{dx} \right) \tag{B-6}$$

and

$$\begin{aligned}
\delta\mathcal{A} &= \varepsilon \int_{\tau_0}^{\tau} d\tau' \int_0^1 dx \left[ \frac{\sigma'(\rho)}{2} r \left( \frac{dH}{dx} \right)^2 + \frac{dH}{dx} \sigma(\rho)\frac{dh}{dx} \right] \\
&= \varepsilon \int_{\tau_0}^{\tau} d\tau' \int_0^1 dx \left[ \frac{\sigma'(\rho)}{2} r \left( \frac{dH}{dx} \right)^2 - H \frac{d}{dx} \left( \sigma(\rho)\frac{dh}{dx} \right) \right]
\end{aligned}$$

One can then eliminate $h$ using (B-6) and this gives

$$\delta\mathcal{A} = \varepsilon \int_{\tau_0}^{\tau} d\tau' \int_0^1 dx \left[ \frac{\sigma'(\rho)}{2} r \left( \frac{dH}{dx} \right)^2 + H \left( \frac{dr}{d\tau'} - \frac{d^2(D(\rho)r)}{dx^2} + \frac{d}{dx} \left( \sigma'(\rho)r\frac{dH}{dx} \right) \right) \right]$$

which (after integrations by parts) can be written as

$$\delta\mathcal{A} = \varepsilon \int_{\tau_0}^{\tau} d\tau' \int_0^1 dx \, r \left[ -\frac{dH}{d\tau'} - D(\rho)\frac{d^2H}{dx^2} - \frac{\sigma'(\rho)}{2} \left( \frac{dH}{dx} \right)^2 \right]$$

where we have used the fact that $H(0, \tau') = H(1, \tau') = r(x, \tau_0) = r(x, \tau)$. As, for the optimal profile $\rho$, the variation $\delta\mathcal{A}$ should vanish for any perturbation $r$, the second equation (136) should be satisfied.

**Acknowledgements**: over the last 25 years, I have greatly benefited from discussions and collaborations in particular with Joel Lebowitz, Eugene Speer, Camille Enaud, Thierry Bodineau, Antoine Gerschenfeld, Tridib Sadhu. I would like to thank them as well Julien Brémont and Samarth Misra who attended my lectures at the Les Houches Summer School and helped me in the preparation of these lecture notes.

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
