# Peer review of "Lecture notes on large deviations in non-equilibrium diffusive systems"

_SciPost Physics Lecture Notes_

## Round 1 · Referee Report · Anonymous (Referee 1) · 2025-7-24

Report

These lecture notes on large deviations in non-equilibrium diffusive systems constitute a pedagogical introduction to the field, together with a nice review of the various results obtained over the last decades. They are well-written and enjoyable to read.

I only have a few minor suggestions to improve the clarity in a few places.

Minor suggestions:

  1. On page 5: it could be useful to define what is meant by "empirical measure", since this term appears here for the first time.

  2. I think it would be useful to briefly define at the beginning of Section 3 what large deviations are. For instance, it is written later on page 12 that the minimum of the large deviation function is zero, which is obvious for someone familiar with the formalism, but otherwise is not so clear.

  3. About Eq. (65): maybe say that it comes from a saddle point calculation of the inverse Laplace transform of (64). That would clarify where the result comes from.

  4. Page 23, Remark 1: it is not so clear why the average current satisfies Fick's law. Naively, one would write

    $$ \langle j(x,\tau) \rangle = - \langle D(\rho(x,\tau)) \rho'(x,\tau) \rangle $$
    but the last step that gives $D(\rho(x,\tau)) \rho'(x,\tau)$ is true because the noise is small, right? Also, up to (115) $\rho(x,\tau)$ denotes the stochastic density, so what does it represent in Fick's law? Should it read $D(\langle \rho(x,\tau) \rangle) \langle \rho'(x,\tau) \rangle$?

  5. On page 38, before Eq. (175) the example of the KMP model is given for an increasing $\sigma(\rho)$. However, it is written at the beginning of the section that the additivity principle that underlies the computations does not apply to the KMP model. Maybe another example could be used for (175) to avoid confusion?

  6. Typos:

  7. page 5, first line: "in in"
  8. page 8, after Eq. (26): "one write"
  9. page 17, before Eq. (84): there is possibly a missing comma after $\mathcal{C}'$
  10. page 22, second line: $(\rho)$ shoul read $f(\rho)$
  11. page 22, before the transport coefficients of the KMP model: "for a system for an isolated system"
  12. page 23, first line: "Fluctuation hydrodynamics" -> "Fluctuating hydrodynamics"?
  13. page 33, in the remark: "known fo"
  14. page 35: in the expression of $H(x,\tau')$ there is a ) missing after $F(x,\tau')$ in the numerator
  15. page 39, in the first remark: there is an additional ) in the expression of $\mu(\lambda)$

Requested changes

Optionally address the minor comments raised in the report.

Recommendation

Publish (surpasses expectations and criteria for this Journal; among top 10%)

---

## Round 1 · Referee Report · Anonymous (Referee 2) · 2025-9-8

Report

These lectures notes give a state-of-the art overview of large deviations in the field of non-equilibrium physics and the macroscopic fluctuation theory. They contain a huge amount of information presented in a very clear and concise manner. Results are explained in an original way and illustrated by simple calculations. The result is both pedagogical and deep. I can only recommend the study of these lectures notes to anyone who aspires to learn the subject from one of the master of the field.

Recommendation

Publish (surpasses expectations and criteria for this Journal; among top 10%)

---

## Editorial Decision

resubmitted